# Halving of Swiss glacier volume since 1931 observed from terrestrial image photogrammetry

Erik Schytt Mannerfelt[1,2], Amaury Dehecq[1,2,3], Romain Hugonnet[1,2,4], Elias Hodel[1,2], Matthias Huss[1,2,5], Andreas Bauder[1,2], and Daniel Farinotti[1,2]

[1]Laboratory of Hydraulics, Hydrology and Glaciology (VAW), ETH Zurich, Zurich, Switzerland
[2]Switzerland Swiss Federal Institute for Forest, Snow and Landscape Research (WSL), Birmensdorf, Switzerland
[3]Univ. Grenoble Alpes, CNRS, IRD, Grenoble INP, IGE, Grenoble, France
[4]LEGOS, Université de Toulouse, CNES, CNRS, IRD, UPS, F-31400 Toulouse, France
[5]Department of Geosciences, University of Fribourg, Fribourg, Switzerland

**Correspondence:** Erik S. Mannerfelt (e.s.mannerfelt@geo.uio.no), Amaury Dehecq (amaury.dehecq@univ-grenoble-alpes.fr)

**Abstract.** The monitoring of glaciers in Switzerland has a long tradition, yet glacier changes during the 20[th] century are only known through sparse observations. Here, we estimate a halving of Swiss glacier volumes between 1931 and 2016 by mapping historical glacier elevation changes at high resolution. Our analysis relies on a terrestrial image archive known as *TerrA*, which covers about 86% of the Swiss glacierised area with 21,703 images acquired during the period 1916–1947 (with a median date of 1931). We developed a semi-automated workflow to generate digital elevation models (DEMs) from these images, resulting in a 45% total glacier coverage. Using the geodetic method, we estimate a Swiss-wide glacier mass balance of –0.52±0.09 m w.e. a$^{-1}$ between 1931 and 2016. This equates to a 51.5±8.0% loss in glacier volume. We find that low elevation, high debris cover, and gently sloping glacier termini are conducive to particularly high mass losses. In addition to these glacier-specific, quasi-centennial elevation changes, we present a new inventory of glacier outlines with known timestamps and complete attributes from around 1931. The fragmented spatial coverage and temporal heterogeneity of the TerrA archive are the largest sources of uncertainty in our glacier-specific estimates, reaching up to 0.50 m w.e. a$^{-1}$. We suggest that the high-resolution mapping of historical surface elevations could unlock great potentials also for research fields other than glaciology.

## 1 Introduction

Glaciers are melting rapidly on a global scale, and constraints on regional- to global-scale volume changes since the 2000s are constantly improving (Gardner et al., 2013; Braun et al., 2019; Zemp et al., 2019; Shean et al., 2020; Hugonnet et al., 2021). Large-scale changes throughout the 1900s are, however, still largely unknown. Recently, modern developments in photogrammetry and structure-from-motion have emerged, making it possible to exploit historical images for glaciological applications (Mertes et al., 2017; Girod et al., 2018; Dehecq et al., 2020). These methods enable the reconstruction of glacier surface geometries for the late 19[th] and early 20[th] centuries, providing either a single snapshot, or in the best of cases, a multi-decadal time series of glacier change (Midgley and Tonkin, 2017; Mölg et al., 2019). These estimates of glacier changes often reveal large discrepancies when compared to model-based approaches (Holmlund and Holmlund, 2019; Belart et al., 2020;

Geyman et al., 2022). To resolve these discrepancies, longer and more frequent observations of glaciers changes are essential, and historical data (e.g. film based images) hold promise to better constrain the response of glaciers to a changing climate.

Glaciers in Switzerland and the European Alps have experienced net retreat since around 1850, albeit at a fluctuating rate, as seen by glaciological mass balance measurements (GLAMOS, 1881–2020; Vincent et al., 2009; Huss et al., 2015; Beniston et al., 2018; Huss et al., 2021). The climate in the 20[th] century was generally unfavourable for glaciers, but mass gains and glacier advances were observed in the 1920s and late 1970s (Huss et al., 2010a, Figure 1C+D). During the last decades, however, an accelerating mass loss trend is seen (Zemp et al., 2019).

While topographic maps resolving Swiss glaciers have been drawn by the Swiss Federal Office of Topography (swisstopo) since the late 19[th] century, today's picture of glacier changes during the last century is largely based on a combination of (i) long-term glacier mass balance series extending back to the 1910s (GLAMOS, 1881–2020), (ii) repeated aerial photogrammetry acquired after the 1960s (Bauder et al., 2007), and (iii) model-based reconstructions (e.g. Huss et al., 2008). Such combined reconstructions are available for selected glaciers, and the results have been extrapolated to the regional scale (e.g. Huss, 2012). In this context, it is important to note that since the uncertainties of glaciological mass balance series are cumulative, they can cause large biases if not corrected with geodetic estimates over longer periods (Cox and March, 2004; Thibert et al., 2008; Zemp et al., 2010, 2013). For Swiss glaciers, such corrections have generally been based on mass changes derived from topographic maps before 1960 and later on from detailed geodetic surveys based on aerial imagery. However, uncertainties in the former datasets are large and sometimes difficult to quantify, especially for the first half of the 20[th] century. Modern reanalysis of the source material for these early maps, thus, holds great potential for further constraining the magnitude and lower the uncertainty of estimates for past glacier changes.

From the First World War to the late 1940s, large parts of the Swiss Alps were surveyed by means of terrestrial (i.e. ground based) photogrammetry. Engineers of swisstopo took photographs from about 7,000 locations distributed across the country, and measured both the terrain and the camera position with a phototheodolite (an angle-measuring device combined with a measuring camera). These surveys, which produced images that are now stored in what is known as the *TerrA* image archive (see Sec. 2.2), served as the basis for the production of the first national topographic maps with a scale of 1:50,000, as well as a set of military maps in the scale 1:10,000.

Here, we digitally process 21,703 terrestrial photographs acquired during the TerrA surveys by using modern photogrammetric methods, and generate digital elevation models (DEMs) of nearly all glaciers in Switzerland (89% by count). The images and DEMs refer to the time span between 1916 and 1947 (with a median year of 1931), and are used to reconstruct the geodetic glacier mass balance from 1931 to 2016 (Figure 1). This is one of the rare regional reconstructions based on digital photogrammetry before the 1950s (e.g. Belart et al., 2020; Geyman et al., 2022), and to the knowledge of the authors, the first of such scope to use terrestrial photographs as the source material.

## 2 Study site and data

### 2.1 Study site

According to the latest Swiss Glacier Inventory (SGI2016; Linsbauer et al., 2021), Switzerland presently contains ca. 1,400 glaciers covering $961.4\pm22\,\text{km}^2$ in total. By area, this is about half of all glaciers in the European Alps (Paul et al., 2020). The Swiss glaciers currently span an elevation range between 1357 and 4599 m a.s.l., with a median elevation of 2913 m a.s.l. and a mean slope of $28°$ (Linsbauer et al., 2021). Both the median elevation and the mean slope are higher than the global average, which are 1308 m a.s.l. and $11°$, respectively (RGI, 2017). This relative steepness makes Swiss glaciers particularly

suited for reconstructions based on terrestrial photogrammetry, as the steepness allows for favourable incidence angles of the photographs. In this study, we consider all glaciers on Swiss territory as long as suitable photographic material is available from the historical surveys (see below).

### 2.2 The "TerrA" terrestrial image archive

The TerrA archive consists of 57,385 images in total, covering most of the mountains in Switzerland and were acquired during

the first half of the 20[th] century. According to the images' approximate viewsheds (produced by swisstopo and distributed together with the images), about 21,703 (38%) of these images cover glaciers.

Images were acquired from high points surrounding the surveyed areas (e.g. summits, ridges, slopes), and stereo acquisition was performed by taking two sets of pictures from locations spaced a few hundred metres apart (Figure 2A). From each location, a panorama of about 4–5 images (Figure 2B) was acquired to increase the viewing angle, with all images being acquired in the

70 same orientation as their stereo-counterparts. About 86% of the glacierised area is covered by at least two historical images from different physical locations, thus setting a constraint to the maximum theoretical coverage of the dataset. The actual coverage is lower, as the incidence angle (i.e. the difference between the angle of the image and the terrain slope) is often low due to the photographs being acquired from the ground. This makes it difficult for feature-matching algorithms to get accurate results, thus lowering the coverage.

The $13\,\text{cm}\times18\,\text{cm}$ terrestrial images contained in the TerrA archive are scanned by swisstopo at a resolution of $21\,\mu\text{m}$, yielding digital images of 53 Mpx. The metadata of each image includes an identifier for which stereo pair it belongs to, the acquisition date, the position, and the viewing direction of the image as determined from field measurements. The photographs used in this study were acquired with 21 individual cameras between 1916 and 1947. Most photographs were acquired in the 1920s and 1930s, with a median year of 1931 and a standard deviation of five years. The cameras were of two different brands

called "Wild" and "Zeiss", which had different focal lengths, image dimensions, and frame geometry (Figure 2C,D).

Within the 2008 "Action Plan for the Preservation of the Spatially Relevant Cultural Heritage of swisstopo", it was decided to preserve the images in the TerrA archive, to scan them all, and to make them available to the public (Ryf and Klöti, 2008; Rickenbacher, 2012). After conceptual preparations, swisstopo started the project in 2013, and announced the release of the dataset in 2018 (swisstopo, 2018). To our knowledge, no scientific publication has used this unique digital dataset to date. The

85 novel use of the imagery proposed here, might thus pave the way for further utilisation of the archive.

## 2.3 Auxiliary data

Our processing workflow (see Sec. 3) requires a modern DEM as a reference for co-registration, as means of validation, and for calculating surface elevation changes with respect to the DEMs generated from the TerrA archive. We use the SwissALTI3D DEM (swisstopo, 2019) for this. This DEM is a 2 m×2 m mosaic of LiDAR (below 2000 m a.s.l.) and digital aerial photography. Over glaciers, acquisition years range from 2007 to 2018, with a median year of 2016. The reported absolute accuracy of the DEM is between 0.1 m and 3 m, depending on the location.

To exclude areas of possible elevation change over the 85-year period, we mask glaciers, perennial snow fields and lakes. To do so, we use (i) the glacier and snow mask that Freudiger et al. (2018) digitised from the "Siegfried maps" made in 1917–1944, and (ii) the swissTLM3D product to mask natural and artificially dammed lakes (swisstopo, 2020).

The TerrA images were previously used to create the first edition of 1:50,000 topographic maps (the so-called "LK50") in the alpine region of Switzerland (swisstopo, personal comm. 2021). We use georeferenced scans of these maps to derive glacier outlines for all glaciers, which are proven here to be concurrent with the TerrA dataset. To digitise glacier outlines from the LK50 map series (see Sec. 3.1.3 for details), we use glacier outlines from the Swiss Glacier Inventory 1973 (SGI1973; Müller et al., 1976) as a template, and follow the same naming convention for glacier identification. The SGI1973 is based on orthophotographs acquired in 1973, and represents the first accurate and complete mapping of Swiss glaciers.

Glacier outlines corresponding to the SwissALTI3D DEM are taken from the SGI2016 (Linsbauer et al., 2021). These are based on detailed mapping performed on orthophotographs with 0.1 to 0.25 m resolution and acquired during the period 2013–2018 (centre year 2016).

Different types of glacier mass balance data covering large parts of the study period are available from long-term monitoring efforts (GLAMOS, 2021). In particular, we use (1) time series of annual glacier-wide mass balance based on in situ observations, available for about two dozen glaciers and partly extending back to the first half of the 20[th] century (Huss et al., 2015, 2021); (2) geodetic mass balances computed by differencing up to 12 DEMs, in turn derived from either topographical maps or aerial photogrammetry (before and after the 1960s, respectively), available for about 50 glaciers at intervals between 3 and 60 years (Bauder et al., 2007; GLAMOS, 2021); (3) modelled time series of annual mass balances covering the period 1900-2021, obtained by updating the results from Huss et al. (2008, 2010a, c), who constrained a distributed daily mass balance model to match the observed glacier mass changes given by (1) and (2); and (4) country-wide, region-specific annual mass balances variations covering the last century, obtained by spatial extrapolation of the observations from (1) and (2), and supported by the modelling of (3) for the period before 1955 (GLAMOS, 2018).

The above data are used for both temporal standardisation of the heterogeneous imaging period, and independent validation of the results (see Methods).

## 3 Methods

In this study, we process 21,703 images semi-autonomously to generate DEMs and measure glacier elevation change between 1931 and 2016. The main steps of the methodology are synthesised in Figure 3, with detailed descriptions provided in the

following subsections. In summary, images are preprocessed first to remove the geometric distortions introduced during digi-tisation (scanning) and to correct for biased position data (Sec. 3.1). Then, DEMs are generated for each stereo-panorama and co-registered to the modern stable terrain (Sec. 3.2). In a third step, the so-derived DEMs are subtracted from the swissALTI3D DEM to obtain elevation change maps for the period 1931-2016. A temporal standardisation is performed to adjust the elevation change rates to said period (Secs. 3.2–3.3.2). Finally, the glacier outlines derived from the digitisation of the LK50 map are used to mask the results and to obtain average elevation change rates and geodetic mass balances (Sec. 3.4).

## 3.1 Image and data preprocessing

### 3.1.1 Fiducial mark detection

We use fiducial marks (small markers on the image frame as fixed references; red squares in Figure 2C+D) to constrain the centrepoint and internal orientation of the imagery. The positions of the fiducial marks are used to calculate a transformation between the pixel coordinates and physical image coordinates. Undesired translations, rotations or scaling-effects introduced during scanning, are removed though a similarity transformation estimated using the Python package "scikit-image" (van der Walt et al., 2014). Each image has four or more fiducial marks along its edges, with a size of about 350 px ($\sim$7 mm). A theoretical minimum of two fiducial marks need to be identified, but more fiducial marks introduce redundancy and allow for exclusion of uncertain fiducial marks. We use a semi-automatic model to identify fiducial marks in the 21,703 images processed in the study. To calibrate the model, we manually identify fiducial marks in 3,395 images ($\sim$162 images per camera). Such a large calibration set was required because the 21 unique cameras in the dataset had slightly or completely different fiducial marks. We then use the manually identified fiducial marks to calibrate the model to do the same on the remaining 81% of the image archive. Using template matching to identify the fiducial marks, the model estimates a similarity transform and validates it using the residual differences between the identified fiducial marks and the modelled ones. We use Random Consensus Sampling (RANSAC) to exclude fiducial marks with residuals of more than 10 px (0.21 mm). The model found four and three fiducials filtered as inliers in 56% and 32% of the images, respectively, with a median manual-to-automatic transform root-mean-square (RMS) difference of 8.61 px (0.18 mm). Model mismatches are largely due to film scratches, damages, or poor contrast in the images, and are easier to identify manually than to account for in a more complex model. Hence, we manually identify the remaining 12% of the images (with fewer than three automatic fiducials) to complement the automatic matches. In seven images, only two fiducials could be identified manually, but they are still used in spite of their higher potential error.

The border of each image needs exclusion to not interfere with the automatic image alignment. We create image frame masks by calculating the median intensity of all images of a given instrument (after applying the geometric transformation), and then segment the median image frame using thresholding.

### 3.1.2 Image position correction

A preliminary analysis of the digitised image metadata revealed spatially correlated biases in the image positions provided by swisstopo. We observe this by calculating the elevation difference between the image position and the SwissALTI3D elevation

on stable ground. While this difference is expected to be the height of the tripod (∼1.2 m; swisstopo, personal comm. 2021), the calculated differences has an absolute bias of 2.66±5.91 (mean±standard deviation) after taking the tripod height into account, and an average horizontal offset of 0.66±4.49 m. A possible explanation for these biases is that camera positions were estimated by triangulation from reference points, which in turn were all positioned relatively to other reference points. Therefore a small positional error could accumulate as the point is further away from the main reference. By sampling slope, aspect and elevation values from the reference DEM at every recorded image position, we estimate the approximate 3D offsets from the image position data to the ground. We then partially correct the systematic component of the biases by averaging them in 1 km×1 km grids, and subtract the gridded offsets from the 3D positions of the images. This correction reduces the average difference between the DEM and the imaging locations to 0.70±5.18 m (0.39±4.15 m horizontally).

### 3.1.3  LK50 map series glacier outline digitisation

We obtain glacier outlines concurrent with the TerrA dataset by digitising outlines on the scanned and georeferenced LK50 map series (in the Swiss CH1903+ / LV95 coordinate system; EPSG:2056) provided by swisstopo. Digitising glacier outlines from orthoimages generated in this study was also a theoretical possibility, but sporadic image coverage and time constraints led to the search for more efficient approaches. These orthoimages are thus only used to draw outlines for 61 sites, which are in turn used to validate the glacier outlines obtained from the LK50 series (see Sec. 3.5.2). The LK50 maps, made in the 1950s, cover all of Switzerland and are based on the TerrA photographs among other data. We manually digitise the glacier outlines of the LK50 maps by modifying the 2,491 outlines (with a mean area of 0.60 km$^2$) of the Swiss Glacier Inventory 1973 as to fit the LK50 map data. This allows for the SGI1973 metadata to be easily retained, and ensures consistency when including or excluding perennial snowfields as well as when drawing ice-divides. We acknowledge that in this way, there is a potential underestimation of the change in the accumulation areas. Indeed, the outlines are only modified where they deviate from the SGI1973 outlines, meaning that minor deviations may be missed. We assume these potential discrepancies to be accounted for in our uncertainty analysis (Sec. 3.5.2). To reduce subjective error, we draw outlines by one person, and validate by another.

### 3.2  Photogrammetry and DEM generation

We process the 21,703 images photogrammetrically in Agisoft Metashape version 1.6.5 in separate subsets (Table 1). Each subset consists of every image acquired with a specific camera over a single year, totalling 113 subsets, with 192 images each on average. We compute a separate camera model for each subset because of varying distortion parameters for each camera, depending on their construction and in which year they were used; differences may relate to the photographer, potential damages, and subsequent repairs the camera might have endured during its lifetime.

Within each subset, we align the stereo-panoramas individually using separate camera models. The ones that successfully align are then merged with a recomputed camera model. The discrepancy between camera models of different years is low for the estimated focal length (0.39% standard deviation on average), while all other distortion parameters (radial, decentering, principal point offset, affinity, and skew) have a spread of 37% on average. The large spread of the distortion parameters could indicate overfitting, but with acceptable tie point residual errors (Table 1), we choose to neglect this issue. The stereo-

panoramas generally overlap with other panoramas in terms of detailed terrain, but the angle difference between extracted features is too large for the built-in feature matcher in Metashape to produce reliable tie points between stereo-panoramas. We therefore perform an alignment between all separate overlapping stereo-panoramas within a subset using Iterative Closest Point (ICP) co-registration of dense point clouds that are constructed from each stereo-panorama. We use the ICP algorithm implemented in the Python package *xdem* (version 0.0.5; xdem contributors, 2021), using core functionality from the *openCV* suite (OpenCV contributors, 2021). We run the co-registration in all areas where the dense clouds overlap, plus a buffer of $\pm15$ m (chosen qualitatively through trial-and-error), to limit the co-registration to reasonable offsets.

Once the stereo-panoramas are aligned to each other, a new camera model is calculated with a bundle adjustment and new dense clouds are generated for each subset individually. We filter the dense clouds by excluding points with a "confidence" of less than or equal to 2. This threshold is chosen qualitatively by visually assessing the effect on noise for a few arbitrarily selected dense clouds. The confidence is a statistic provided by the Metashape software, but information on how it is calculated is unavailable. After the DEMs are generated from the resultant dense clouds, and after orthoimages are generated using the DEMs, we co-register the DEMs to the SwissALTI3D DEM (Figure 4A+B). This is done using the same ICP co-registration method as in the previous step, but with the modern DEM as a reference outside of the unstable terrain mask. We use the ICP transforms to correct the positional and rotational errors of both the historical DEMs and orthoimages. Finally, elevation change maps are generated by subtracting the DEMs from the reference SwissALTI3D DEM.

## 3.3 Post-photogrammetric processing

### 3.3.1 Temporal standardisation

Because both the historical and modern elevation data set were acquired over several years (1916–1947 and 2007–2018, respectively), it is necessary to standardise the periods of observation for the evaluation and interpretation of the results.

We adjust each elevation change map to the median years 1931 and 2016 based on an annual dataset of observed mass balance variations that is specified for the major hydrological basins of the Swiss Alps (point "(4)" in Sec. 2.3). Regional anomalies in mass balance from the reference period 1961–1990 have been derived by spatially extrapolating all available series acquired using the direct glaciological method (between 8 and 20 glaciers; Huss et al., 2015; GLAMOS, 1881–2020). Before 1955, modelled mass balance variations for four glaciers were used (Huss et al., 2008). For each pixel, we calculate the ratio between the cumulative mass balance of the reference period 1931–2016 and the cumulative mass balance over the actual period of observation (Equation 1). The elevation change rates over glacierised areas are then multiplied by this ratio to obtain the temporally standardised elevation change rate $\left[\frac{dH}{dt}\right]_{1931-2016}$:

$$\left[\frac{dH}{dt}\right]_{1931-2016} = \left[\frac{dH}{dt}\right]_{t_0-t_1} \times \frac{B(2016) - B(1931)}{B(t_1) - B(t_0)}, \tag{1}$$

where $t_0$ and $t_1$ are the years of acquisition of the TerrA photograph and the SwissALTI3D DEM, respectively, and $B(t)$ is the representative cumulative mass balance at year $t$ for the given region.

### 3.3.2 Mosaicking and interpolation

We successfully generate 1,907 elevation change maps, which we mosaic by averaging the temporally standardised elevation changes estimated at each pixel in a 5 m×5 m grid. We exclude three out of the 1,907 elevation change maps from the mosaic due to extreme median values (<−350 m or >100 m). Regionally, 55% of the total glacier area is missing from the elevation change mosaic due to incomplete image coverage and processing shortcomings, so an interpolation approach is required (Figure 6). Gaps are spread over glacier extents throughout their elevation ranges, and we therefore chose to harness hypsometric interpolation approaches.

We use a modified version of the regional hypsometric approach (referred to as "global" in McNabb et al., 2019) where each glacier elevation change is normalised to its elevation range, i.e. scaling both elevation and elevation change from 0 to 1 for each glacier in 5$^{th}$ percentiles (Huss et al., 2010b). More specifically, we use the approach as implemented in the Python package *xdem*. This approach relies on elevation changes estimated at different elevation bins, and aims at capturing the glacier-wide hypsometric signal. Consequently, the quality of the interpolation deteriorates with decreasing spatial coverage. For our application, we choose this approach where >20% of the glacier area is covered (91.1% of all glaciers by area meet this criterion). For glaciers that do not meet this condition, we apply a simple regional hypsometric approach, in which gaps are replaced by the mean regional value within the same elevation band.

### 3.4 Glacier-specific and regional elevation change

We estimate glacier-specific elevation changes and geodetic mass balances for each SGI2016 outline individually. Then, we aggregate glacier-specific estimates to obtain regional-scale estimates.

For each individual glacier, we estimate the mean elevation change $\frac{dH}{dt}$ by averaging both the observed and interpolated elevation changes that fall within the historical glacier outline:

$$\frac{dH}{dt} = \frac{\sum \left[\frac{dH}{dt}\right]_p}{N_p}. \tag{2}$$

Here, $\left[\frac{dH}{dt}\right]_p$ is the elevation change rate of a given pixel $p$, and $N_p$ is the number of pixels within the glacier outline. We then estimate the mean volume change rate $\frac{dV}{dt}$ by:

$$\frac{dV}{dt} = \frac{dH}{dt} \times A_0, \tag{3}$$

where $A_0$ is the glacier area as given by the historical glacier outline (referring to ca. 1931).

By applying a volume-to-mass conversion factor of $\rho_{\Delta V} = 850 \, \text{kg m}^{-3}$ (Huss, 2013), we estimate the specific-glacier mass balance rate $\dot{B}$ using:

$$\dot{B} = \rho_{\Delta V} \times \frac{\frac{dV}{dt}}{(A_0 + A_1)/2}, \tag{4}$$

where $A_1$ is the glacier area as given by the SGI2016.

Similarly, we estimate the total glacier mass change rate $\dot{M}$ from the volume change and the volume-to-mass conversion factor:

$$\dot{M} = \rho_{\Delta V} \times \frac{dV}{dt}. \tag{5}$$

## 3.5 Uncertainty analysis

We identify multiple sources of error that propagate to our estimates of glacier volume and mass change. These sources include stochastic elevation change measurement errors, correlated short- and long-range DEM distortions, error in glacier outlines, the temporal standardisation, hypsometric interpolation, and the volume-to-mass conversion. We consider these sources separately, and combine them into estimates of uncertainty for elevation, volume, and mass change. Throughout the text, uncertainties are within a 95% confidence interval.

To propagate uncertainties from the pixel-scale to the glacier-scale and from the glacier-scale to the regional scale, we estimate the spatial ranges at which errors are correlated. Where relevant, we perform this propagation by estimating the standard error with a number of effective samples ($N_{\text{eff}}$) that account for spatial correlations. We derive the number of effective samples by numerically integrating a sum of variogram models fitted to our empirical variograms within a circular area of the same size as the glacier (Figure 4; Hugonnet et al., 2021). This is a generalisation of the approach by Rolstad et al. (2009). We use the implementation in the Python package *xdem* based on spatial statistics of the Python package scikit-gstat (Mälicke, 2022).

### 3.5.1 Mean elevation change uncertainty

Multiple intermediate steps in the photogrammetric pipeline can distort DEMs and orthoimages. These include image scanning artefacts, camera model uncertainties, image noise, image location and orientation error, as well as subsequent methodological shortcomings (Dehecq et al., 2020). We quantify the uncertainty comprising these errors by comparing terrain assumed to be stable in our elevation change estimates. Stable terrain excludes ice, perennial snow patches (Freudiger et al., 2018) and lakes. Dams created between the 1920s and 1980s are especially important to exclude, as they otherwise introduce unwanted positive changes of up to >50 m. While other factors such as landslides, vegetation change and buildings may also affect stable terrain, visual inspection showed no obvious signs for their presence.

For DEMs generated by satellite or airborne photogrammetry, elevation change errors often vary with terrain aspect, slope, or curvature (Toutin, 2002; Hugonnet et al., 2021). Here, we find no dependency of the uncertainty of elevation change on these terrain attributes. We suspect that this is the result of the large number of DEMs that is used in our mosaic. Indeed, DEMs derived from terrestrial images might be affected by such attribute-dependent errors, but since the individual DEM tiles are derived independently from an independent set of images, these should cancel out when aggregated over several glaciers. We thus assume that our elevation uncertainty is stationary in space, and estimate it by using the Normalised Median Absolute Deviation (NMAD) of the elevation change estimates on stable terrain.

We account for spatial correlation of elevation change uncertainty by fitting a double-spherical variogram model to the empirical variogram of elevation changes (Figure 4D+E), thereby modelling short- and long-range correlations (respectively in stable terrain residuals, panel E, and interpolation errors, panel D). We then estimate the uncertainty in the mean elevation change rate $\sigma$ by circular integration of the sum of variogram models over the glacier area $A$ (Rolstad et al., 2009; Hugonnet et al., 2021):

$$\sigma = \frac{2\pi}{A} \int_{r=0}^{\sqrt{A/\pi}} [\sigma_p - \gamma(r)]\, dr = \frac{\sigma_p}{\sqrt{N_{eff}}} \tag{6}$$

where $\sigma_p$ is the NMAD of elevation differences on stable terrain, $\gamma(r)$ is the sum of variogram models with spatial lag $r$ and $N_{eff}$ is the number of effective samples.

### 3.5.2   Glacier outline uncertainty

We identify three major sources of uncertainties from the glacier outlines drawn from the LK50 maps: the georeferencing error of the map, the timing uncertainty of the used topographic data (which is unknown, but was assumed to largely be the TerrA dataset), and the errors in manual delineation (including the subsequent digitisation from the map) of a glacier front. The temporal concurrency of the LK50 map with the TerrA imagery particularly needs validation, as the extent of which data from other times was used is unknown. In the case where the outlines are spatially consistent with the TerrA dataset, the concurrency assumption is verified.

We evaluate these three uncertainty sources combined by digitising sparse outlines at 61 sites from orthorectified versions of the photographs generated in the photogrammetric pipeline. We then compare their general agreement with the LK50 glacier outlines. Outlines drawn from orthoimages will have a perfect relative georeferencing to the DEMs, and should thus only differ from the LK50 outlines due to the subjective nature of glacier delineation. For this comparison, we draw lines from the centroid of each glacier toward arbitrary points on the digitised outline and measure the distance to each LK50 and orthoimage-derived outline intersection. We use the difference between the two as a metric of the variability in spatial extent (Figure 5B).

We find a resultant median length difference of 7.9 m; the LK50 outlines are generally less extensive, possibly due to manual delineation differences in including or excluding glacier-marginal snow cover, with an NMAD of 28.06 m. Finally, in order to propagate these delineation differences into a source of elevation changes uncertainties, we first dilate (i.e. enlarge) and erode (i.e. shrink) the mask by the systematic difference (7.9 m). We then estimate the glacier outline uncertainty by computing the average difference between the initial mask and the dilated or eroded masks:

$$\sigma_{area} = \frac{\left| \overline{\left[\frac{dH}{dt}\right]_D} - \overline{\left[\frac{dH}{dt}\right]_E} \right|}{2}, \tag{7}$$

where $\overline{\left[\frac{dH}{dt}\right]_D}$ is the mean elevation change rate within the dilated mask, and $\overline{\left[\frac{dH}{dt}\right]_E}$ is the mean elevation change rate within the eroded mask.

### 3.5.3 Interpolation uncertainty

We estimate the uncertainty in the interpolation approach by artificially introducing gaps in the gap-free elevation changes of Hugonnet et al. (2021). These elevation changes are based on stereo-correlation of optical satellite imagery from the Advanced Spaceborne Thermal Emission and Reflection Radiometer (ASTER), which cover the entire study area, and refer to the period 2000–2020. We interpolate the introduced gaps using our approach, and study the difference between the interpolated ASTER elevation changes and the original ASTER elevation change estimates. We use this difference solely to compute the spatial correlation of uncertainties. For this, we fit a double-spherical variogram model to the empirical variogram of the above difference in elevation change. We thereby measure the spatial correlation range of the error of interpolated elevation changes (Figure 4D). In order to scale the variance of our interpolation uncertainty $\sigma_{int}$ correctly, we did not use the NMAD of ASTER differences which have a different absolute magnitude, and could be biased by a lesser vertical precision. Instead, we interpolate the TerrA elevation changes for all pixels and then use the NMAD of the difference between the original and interpolated elevation change estimates.

We find strong correlations of interpolated elevation change errors at pixel scales, which fully decorrelate at a spatial lag of approximately 2,100 m. This indicates that errors are strongly correlated at the glacier scale, but are largely independent between glaciers of a given region. We thus propagate our interpolation uncertainties from pixel to glacier or region using Equation 6. We then add a 0.01 m a$^{-1}$ 1-$\sigma$ symmetrical uncertainty to conservatively account for the median bias found in the difference between the interpolated and original elevation changes from ASTER.

### 3.5.4 Temporal standardisation uncertainty

We perform our temporal standardisation using modelled mass balance estimates, whose uncertainties accumulate over time. The reported uncertainty for one year's specific mass balance ($\sigma_B$) based on in situ glaciological observations is approximately 0.2 m w.e. a$^{-1}$ (Zemp et al., 2013). We derive the uncertainty in the temporal standardisation by standard propagation of independent uncertainties:

$$\sigma_{time}^2 = \left[\frac{dH}{dt}\right]^2 \times \left[\frac{2\sigma_B^2}{B(2016) - B(1931)} + \frac{2\sigma_B^2}{B(t_1) - B(t_0)}\right] \times \left(\frac{B(2016) - B(1931)}{B(t_1) - B(t_0)}\right)^2, \tag{8}$$

where $B(t)$ is the cumulative mass balance for year $t$, and where the standardisation is carried out for the time period $t_0$–$t_1$.

### 3.5.5 Total elevation change uncertainty

We calculate the total glacier-specific and regional uncertainty in mass balance by the squared sums of all error components, accounting for the number of effective samples ($N_{\text{eff}}$) derived from the variograms where appropriate:

$$\sigma_{total}^2 = \left(\frac{\sigma_{topo}}{\sqrt{N_{topo}}}\right)^2 + \sigma_{time}^2 + \sigma_{area}^2 + \left(\frac{\sigma_{int}}{\sqrt{N_{int}}}\right)^2 + \left(\frac{|\dot{B}| \times \sigma_{\rho_{\Delta V}}}{\rho_{\Delta V}}\right)^2 \tag{9}$$

Here, $\sigma_{topo}$ is the NMAD of the stable terrain difference, $N_{topo}$ is the number of effective samples calculated from the stable terrain variogram model (Figure 4E) and using Equation 6, $\sigma_{int}$ is the NMAD of the difference between the artificially removed and interpolated pixels, $N_{int}$ is the number of effective samples calculated from the interpolation error variogram model (Figure 4D, Equation 6), $|\dot{B}|$ is the absolute average mass balance rate for the glacier or region, and $\sigma_{\rho_{\Delta V}} = 60\,\text{kg m}^{-3}$ is the uncertainty of the factor used for volume-to-mass conversions (Huss, 2013).

## 4    Results

For the period 1931–2016, our results indicate an average glacier volume loss of $0.73\pm0.08$ km$^3$ a$^{-1}$, and a mean annual specific mass balance of -0.52$\pm$0.09 m w.e. a$^{-1}$ (Table 2). With a total Swiss glacier volume of 58.7$\pm$2.5 km$^3$ in 2016 (Grab et al., 2021), this corresponds to a loss of 0.86$\pm$0.08% per year compared to the 1931–2016 average volume. Integration of these changes over the considered period provides a total glacier volume loss of 51.5$\pm$8.0% since 1931 (0.15$\pm$0.03 mm sea level equivalent assuming 361.8 Gt mm$^{-1}$; Hugonnet et al., 2021). The total glacier area concomitantly reduced by 6.2$\pm$0.8 km$^2$ a$^{-1}$, corresponding to an area loss of 35.6$\pm$6.5% (the total glacier area was 1492$\pm$68 km$^2$ in ca. 1931, and 961$\pm$22 km$^2$ in 2016).

We analyse the spatial variability in mass balance over a regular grid of 30 km$\times$30 km (Figure 7). The results show relatively high spatial variability, as shown by the ca. 1.5$\times$ larger mass loss rates in the Aletsch region compared to the Southern Valais Alps. On the same grid, we observe a minimum regional loss of 0.32 m w.e. a$^{-1}$ in the south west, and a maximum loss of 1.02 m w.e. a$^{-1}$ in the north east (only cells with more than five glaciers are considered).

Analysing the regional mass balance as a function of easting and river basin (Figure 8) reveals a clustered elevation dependency for individual basins. In general, glaciers in the central part of the Rhone basin have a higher median elevation and lower mass loss compared to other basins. We also observe a Pearson correlation coefficient of $r = 0.42$ between glacier mass balance and median elevation (Figure 9A), indicating that lower glaciers appear to experience higher mass loss rates. From these two results, we can conclude that the regional variability in mass balance across the Swiss Alps is largely influenced by glaciers' median elevation.

Weaker correlations with other morpho-topographic characteristics are detected as well. For example, we observe a correlation between glacier mass balance and the mean slope of the lowest 10[th] percentile of elevation ($r = 0.28$; "terminus slope"; Figure 9C), and a similar negative relationship with modern fractional debris cover ($r = -0.28$; Figure 9D). This indicates that glaciers with flatter termini or higher debris-cover fraction tend to experience higher mass loss rates than their steeper and ice-free counterparts. These findings support the results from studies performed with smaller subsets of Swiss glaciers (Huss, 2012), over shorter time spans (Fischer et al., 2015), in different geographical regions (Brun et al., 2019; Geyman et al., 2022), or based on area-change data (Linsbauer et al., 2021).

## 5    Discussion

### 5.1    Comparison with previous glacier-wide estimates

The current consensus on the magnitude of Swiss glacier changes over the 20[th]century stems from (i) the work by Bauder et al.
(2007), who derived glacier volume changes from the digitisation of topographic maps and the evaluation of repeated aerial
photogrammetrical surveys for 19 large glaciers; and (ii) the combination of the analysis by Bauder et al. (2007) with direct
glaciological measurements and modelling, extended to 39 glaciers (Huss et al., 2010a, c; GLAMOS, 1881–2020). Since the
topographic maps relied on the TerrA surveys, comparing our results with these previous estimates is particularly relevant. We
thus extract the cumulative mass balance for each of the 39 glaciers considered in previous studies, and compare these results
to our estimates (Figure 10A). We find generally good agreement at the scale of the individual glaciers (r=0.71), although the
results from the previous studies (Bauder et al., 2007; Huss et al., 2010a) indicate a mass loss rate that is, on average, 0.08 m
w.e. a$^{-1}$ smaller than ours.

For Grosser Aletschgletscher (location shown in Figure 6), we additionally compare the map data digitised by Bauder et al.
(2007) to the DEM mosaic we generated for 1926 and 1927 (Figure 10B–E). The mean thickness change derived from the map
is –56 m (with an unknown uncertainty) while the results of our study suggest a change of –72±10 m (Figures 10B and 10C,
respectively). Comparing the two DEMs reveals an elevation-dependent difference (Figure 10D-E): the results generally agree
in the accumulation area, but differ considerably at the low glacier elevations. We attribute this difference to either (i) poor
georeferencing of either datasets, (ii) temporal inconsistencies in the map data, which might not uniformly represent the year
1927, or (iii) a combination of these factors. Also note that for being consistent with Bauder et al. (2007), the above values are
obtained by excluding the tributary Mittelaletschgletscher. Before 1969, the latter was confluent with Grosser Aletschgletscher
(GLAMOS, 1881–2020, this is also reflected in the LK50 outlines). Similar outlines inconsistencies at other glaciers might
contribute in explaining differences between the two studies as highlighted on Figure 10A.

The above comparison also reveals a potential shortcoming of our interpolation approach near the glacier margins. Indeed,
the proximity to glacier margins was not considered, meaning that changes can be overestimated at these specific locations.
The problem arises from the fact that our gap-filling procedure is based on the multiplication of the average ice-thickness
change rate observed at a given elevation with the length of the considered time period (i.e. 1931-2016, on average). This
results in a total, elevation-dependent ice thickness change that is added back to the SwissALTI3D DEM. For a given location,
however, the actual surface elevation in 1931 might have been smaller, especially if the location has become ice-free before
2016. The latter case is most often encountered near the glacier margins, which explains the pattern of differences observed in
Figures 10D and 10E.

While the above shortcoming is inherently accounted for in our uncertainty analysis, we suggest that future studies could
improve the interpolation technique used here. Auxiliary information on the timing at which a given location has become ice
free might help with that. Assuming that the uncertainty of the maps used by Bauder et al. (2007) is similar or larger than in
our study, however, the 95% confidence intervals of the mean thickness change overlap.

## 5.2 Comparison with the regional estimate of Huss (2012)

To date, the only Swiss-wide estimate of glacier volume changes over a century is provided by Huss (2012), who extrapolated mass balance time series available for 50 glaciers by using a multi-parameter regression. The study refers to the period 1900–2011, and estimated a total volume loss of $-42.1$ km$^3$, corresponding to a mean surface mass balance of $-0.28$ m w.e. a$^{-1}$. To make these results comparable to our study, we updated the estimate to the period 1931–2016 by using the same methodology as in the study by Huss (2012). This results in a Swiss-wide volume loss of $-44.8$ km$^3$, which is 28% smaller than our estimate. The reasons behind this discrepancy might be twofold: for one, our interpolation scheme may overestimate thinning over retreating glacier areas (cf. Sec. 5.1), for another, the two studies are based on two different glacier inventories (we rely on the SGI2016 while Huss (2012) relied on an inventory from 2003). Here below we discuss these two possible explanations in more detail.

To evaluate the influence of our interpolation scheme, we estimate the range of the associated bias in two ways. (1) In the first way, we simply assume a constant bias of $-0.08$ m w.e. a$^{-1}$ for our estimate, i.e. 15% of the total volume change (see Sec. 5.1). This results in a correction of $+9.4$ km$^3$. Note, however, that such a correction implies the strong assumption that map-based observations are error free, which seems difficult to defend. (2) In the second way, we consider that the interpolated thinning rates over now deglacierised pixels could be overestimated by a factor 2. This factor 2 results from the assumption of a constant thinning rate and a constant glacier retreat. In such a case, indeed, the average thinning experienced by the area that deglacierised between 1931 and 2016 would be the average between zero (i.e. the thinning experienced by pixels at the glacier boundary in 1931) and the maximum thinning (i.e. the thinning experienced by pixels at the glacier boundary in 2016), while in our method all interpolated values are set to the maximum thinning (this is because the year at which a given pixel deglacierised is unknown). From the (i) deglacierised area (Table 2), (ii) average elevation change over deglacierised pixels with interpolation ($-50.8$ m), and (iii) fraction of deglacierised interpolated pixels (0.44), we estimate a correction of $+5.8$ km$^3$. The two ways thus indicate that our interpolation scheme could introduce a negative bias of 5.8–9.4 km$^3$. Since the exact bias is unknown, however, we prefer to provide the uncorrected estimates and to account for the bias in our uncertainty assessment.

In relation to the different glacier inventories used by Huss (2012) and by us, we note that many small glaciers have disappeared between 1931 and 2003 (i.e. the year of the inventory used by Huss (2012)). These glaciers could not be accounted for in the Huss (2012) estimate. To quantify this effect, we identify all disappeared glaciers, i.e. all glaciers polygons in the 1931 inventory that do not overlap with any polygons of the 2016 inventory. We find that such glaciers account for 68 km$^2$ of the area loss and for 2.3 km$^3$ of the volume loss since 1931. The latter corresponds to 3.6% of the loss estimated for the whole sample. The mean thinning rate for these glaciers is roughly 25% lower than the regional average, which can be explained by the fact that the thinning is bounded by the initial ice thickness. The contribution of 3.6% from disappeared glaciers is slightly smaller but comparable to the 5–6% contribution estimated by Parkes and Marzeion (2018) for glaciers at the worldwide scale.

When combining the above considerations, i.e. when applying a possible correction of between 9.4 and 5.8 km$^3$ because of the interpolation method and when accounting for a 2.3 km$^3$ difference because of disappeared glaciers, our estimated volume

loss would come to lie in the range of 50.7–54.3 km$^3$. This is still ca. 15% larger than the estimate by Huss (2012) but reconciles the two estimates within their respective uncertainties.

## 5.3 Main drivers of variability in glacier mass loss

To better understand the drivers of glacier mass balance, we perform a multivariable regression analysis identical to Huss (2012). When doing so, we estimate the fraction of variance explained by each explanatory variable by calculating the difference in the coefficient of determination ($R^2$) between the case in which the given variable is included and the case in which it is not. Table 3 shows that the highest portion of variance is explained by the median glacier elevation, followed by the glacier area. The glaciers' easting and northing, as well as the mean glacier aspect (tested for consistency with Huss, 2012) only explain a negligible fraction of the variance and are therefore excluded from further analysis.

These findings are similar to the ones by Huss (2012), except for the fact that median elevation has a higher importance than glacier area in our study (Huss, 2012, found the opposite) . The difference may be due to the different sample sizes (50 and 2491 glaciers, respectively), the different glacier outlines (which directly affect the glacier area), or the different time intervals. While the importance of the median elevation is clearly reflected in a direct correlation with the determined mass change rates (Figure 9A), the importance of glacier area is more nuanced and only a weak correlation is observed (Figure 9B). This could imply a co-variance between glacier area and another parameter (such as the median elevation) which, in combination, would help explaining the observed variance in mass balance.

We attribute the correlation between median elevation and glacier mass loss to differences in regional snow accumulation: as the forcing resulting from air temperature is similar throughout the Swiss Alps (Isotta et al., 2019), we hypothesise that variations in median glacier elevation (Figure 8) are driven by differences in precipitation totals that are reduced in the inner-alpine regions of the Valais and Eastern Switzerland GLAMOS (1881–2020). A glacier located in a region with drier – or more continental – climate will thus experience a higher median elevation. In turn, precipitation totals are known to determine altitudinal mass balance gradients, and hence the sensitivity of glacier mass balance to changes in air temperature (Oerlemans and Reichert, 2000; Rasmussen and Andreassen, 2005). In summary, regions with smaller precipitation are characterized by higher glacier median elevation and, hence, reduced climate change sensitivity, resulting in less negative mass balance compared to regions with high precipitation totals.

The single observation time period in our study (1931–2016) prevents us from analysing the climatic factors (e.g., temperature, precipitation) that control the temporal variability in glacier mass balance. Such an analysis was already performed by Huss et al. (2010a), who used annual mass balance series and found that the multi-decadal mass balance variability is significantly anticorrelated to the Atlantic Multidecadal Oscillation. By leveraging different data sets, e.g., from repeated aerial surveys or satellite observations, it would be possible to obtain more frequent observations (Bauder et al., 2007; Rastner et al., 2016; Mölg et al., 2019). This could help constraining the multi-decadal variability of glacier mass changes, but is understandably beyond the scope of our study.

## 5.4 Advantages and challenges of digital reanalysis

Reanalysis of the source material is a clear step forward compared to the map-based analyses that were available so far (cf. Section 5.1). This is because the generation of topographic maps requires considerable interpretation, and sometimes qualified guesswork to connect points of known elevation with continuous contour lines. The two factors make for a fuzzy border between actual data and subjective inter- or extrapolation. Comparisons between glacier extents obtained from digitised maps and modern digital photogrammetric reconstructions, show significantly reduced biases for the latter (e.g. Koblet et al., 2010; Midgley and Tonkin, 2017; Geyman et al., 2022). In addition, many maps are temporally inconsistent. This is because maps are asynchronously updated with newer data, which are added to older maps during successive revisions. Problematic in this respect is the fact that, at least for Switzerland, the exact year of the data used during such revisions is mostly unknown (Fischer et al., 2015). This issue has been solved for maps referring to the second half of the 20th century, where precisely dated and complete surveys based on aerial photogrammetry have become available. This is an important step forward, especially for glaciers with long-term monitoring programmes (Bauder et al., 2007).

While the temporal consistency of our results is warranted in general, some uncertainty is introduced through the temporal standardisation required for individual glaciers. For individual glaciers, this uncertainty is generally below $0.09\,\mathrm{m\,w.e.\,a^{-1}}$ (see Sec. 3.3.1 and Figure 4C). Such temporal inhomogeneity of country-wide results is inherent to any analysis that is based on terrestrial surveys. Indeed, the time required to acquire such data will always lead to a temporal spread in the time of acquisition. Although this problem is decreasing in importance for modern surveys thanks to a wider image swath, the analysis of historical photographs requires careful consideration of the issue. We suggest that temporal standardisation, in line with (or better than) our study, is a fundamental step in post-processing and interpreting the results derived from historical surveys – especially terrestrial ones.

Future improvements of our workflow may help further increasing the spatial coverage of the dataset and reducing co-registration errors. While our Swiss-wide total specific mass balance uncertainty is low ($0.09\,\mathrm{m\,w.e.\,a^{-1}}$), uncertainties at the scale of individual glaciers are in the same order of magnitude as (and sometimes higher than) the magnitude of the change ($0.01$–$0.62\,\mathrm{m\,w.e.\,a^{-1}}$, with a median of $0.17\,\mathrm{m\,w.e.\,a^{-1}}$). The interpolation necessary over 55% of the analysed area was the largest contributor of uncertainty in our analysis ($0.00$–$0.39\,\mathrm{m\,w.e.\,a^{-1}}$ for individual glaciers; Figure 4C). Higher coverage, or a more accurate interpolation method, may reduce this. The TerrA dataset has a theoretical maximum glacier coverage of ca. 86%, meaning that only about half of the theoretical maximum coverage was realised in our reconstruction. Newer photogrammetric methods, especially those rooted in the machine learning domain (e.g. Mildenhall et al., 2020; Zheng et al., 2020; Sarlin et al., 2020), can help improve feature description or depth reconstruction. This may increase the coverage of the resulting datasets compared to the methods currently implemented in Agisoft Metashape. In this respect, we attempted to replace the feature matcher with ASIFT (Morel and Yu, 2009), but long processing times and a general lack of robustness to image noise led us to abandon the idea.

Concerning the TerrA dataset, it is remarkable that reanalyses such as presented here are possible at all for a dataset that is almost hundred years old, and in some cases even older. This possibility is only provided by the systematic, thorough, and

meticulous archiving that was performed at that time, and by the preservation efforts that were invested since then. The Swiss Federal Office of Topography deserves credit for this. In this context, it is interesting to note that what have now become known as the "FAIR Guiding Principles for scientific data" (Wilkinson et al., 2016) were respected over the course of almost a century. Indeed, the FAIR principles require the data to be findable (F), accessible (A), interoperable (I), and reusable (R), and all of these criteria are met for the TerrA archive: the images are easily found and accessed via swisstopo's geodata portal[1], are made interoperable by being distributed in a commonly accepted digital data format, and are reusable thanks to the rich and self-explaining metadata that complement the individual images. Similar databases exist in many other countries including, and not limited to, the United States of America[2], France[3] and Sweden[4]. We encourage data holders and archivists alike to follow such examples, and hope that our study can be an example demonstrating the added value of such long-term archiving efforts.

## 6 Conclusions

In this study, we utilise 21,703 terrestrial photographs from the "TerrA" archive recently made accessible by the Swiss Federal Office of Topography to quantify glacier change over the entire Swiss Alps since 1931. We take advantage of modern photogrammetric techniques and the software Agisoft Metashape to re-process the scanned images in a semi-automated fashion. We use the procedure to reconstruct the ca. 1931 topography for 45% of the glaciers in the region. We estimate a Swiss-wide glacier mass balance of $-0.52\pm0.09$ m w.e. $a^{-1}$ and an area reduction of $6.2\pm0.8$ km$^2$ a$^{-1}$ between 1931 and 2016. This translates to a halving of glacier volume, and a reduction in areal cover by a third. Our results indicate a strong spatial variability in glacier thinning, with glaciers in the north east losing mass twice as rapidly as in the south west of Switzerland. This variability is partially explained by the fact that mass losses are found to be pronounced for glaciers at a lower median elevation, with more gently-sloping termini, and with a high present-day debris-cover fraction.

While our approach is successful in obtaining regional estimates, observations at the local scale are still uncertain for the analysis of individual glaciers. Uncertainties in our estimates are particularly dominated by a fragmented spatial coverage and the temporal heterogeneity of the TerrA acquisitions, which account for 33% and 32% of the uncertainties, respectively. In particular, we identify that the gap-filling, based on the regional hypsometric approach, could overestimate mass loss over deglacierised areas. The exact bias is unknown but it could reach up to 0.08 m w.e. $a^{-1}$, or 15% of the estimated volume loss. Among the main challenges of dealing with such terrestrial archives are (i) the limited extent of the observed area, hence reducing the amount of stable terrain available for DEM co-registration, (ii) the low incidence angle of the images that causes geometric distortions, and (iii) the long time span between acquisitions. While all these issues are inherent to terrestrial images, the first two challenges call for improved image processing methods to find correspondences between images acquired

[1]https://www.swisstopo.admin.ch/en/geodata/images/terrestrial.html (last accessed 01/07/2022)
[2]https://earthexplorer.usgs.gov (last accessed 01/07/2022)
[3]https://remonterletemps.ign.fr (last accessed 01/07/2022)
[4]https://www.alvin-portal.org, (last accessed 01/07/2022)

with strongly differing view-angles if improved small-scale accuracy is sought. The combination of remote sensing, in situ observations and modelling was the key in solving the third challenge and providing a temporally consistent estimate.

Ultimately, our results validate numbers that were only previously inferred by extrapolation from sparse data. In a period of rapid increase in air temperature, regional mass balance data over a time span of almost one century are essential to accurately understand how glaciers respond to changes in climate. The reconstructions of this study, among other recently published datasets of regional scope, may mark a milestone for further understanding of past, current and future glacier change. The approaches developed here are expected to offer an equal potential for evaluating other historical datasets with modern tech-

nology.

*Acknowledgements.*  We would like to thank swisstopo for acquiring, archiving, scanning and distributing the TerrA images. This work would not have been possible without over a hundred years of efforts. We are particularly grateful to Philip Joerg and Matthias Zesiger for providing additional details on the images metadata and acquisition. We are grateful to Roxana Zehtabchi for early developments of the data processing workflow. We would like to thank Timothée Produit and Adrien Gressin (HEIG-Vaud) for sharing initial TerrA viewsheds and for discussions

on the TerrA archive. This project was funded by the Federal Office of Meteorology and Climatology MeteoSwiss in the framework of GCOS Switzerland.

*Code and data availability.*  The source code of the photogrammetric analysis is available at: https://github.com/VAW-SwissTerra/SwissTerra. The code for post-processing, uncertainty, and mass balance is available at: https://github.com/VAW-SwissTerra/terradem. The Python package "xdem" was partly developed for this project, and is available at: https://github.com/GlacioHack/xdem.

All intermediate and output data for this analysis is available at: https://doi.org/10.5281/zenodo.6675911.

*Author contributions.*  AD and DF conceived the study. EM wrote the study code and performed the analyses with main input from AD, as well as RH and DF. EM made the figures and wrote the majority of the manuscript with inputs from AD, RH, MH, EH and DF. EM and EH created the LK50 glacier outline inventory. AD and RH contributed with *xdem* code and analyses. EH, MH and AB contributed with data to the analysis. All authors contributed to the manuscript text.

*Competing interests.*  The authors report no competing interests.

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

**Table 1.** Photogrammetric processing parameters. The "High" alignment quality and "Ultra High" dense cloud quality means using the full image resolution for each respective step. "Mild" (1/3 strength) dense cloud filtering is a proprietary Metashape filtering setting. Rows with "*" represent averages for each individual stereo-panorama.

| | |
|---|---|
| Fiducial matching | |
| Manually identified fiducials | 3,395 |
| Automatically identified fiducials | 19,123 (81%) |
| Automatic vs. manual RMSE | 8.61 px (0.18 mm) |
| Position correction | |
| Initial position offset | 2.66±5.91 m |
| Corrected position offset | 0.70±5.18 m |
| Agisoft Metashape | |
| Mean subset (chunk) size | 192 images (12 stereo-panoramas). |
| "Zeiss" camera focal lengths | 194/195 mm |
| "Wild" camera focal lengths | 150–240 mm |
| Alignment parameters | High quality, 4,000 tie points, 40,000 key points. |
| Mean tie point count* | 6937 |
| Mean tie point reprojection RMSE | 0.94 px |
| Reference accuracies | ±2 m in position / ±1° in rotation. |
| Dense cloud parameters | Ultra High quality, Mild filtering. |
| Mean dense cloud point count* | 23,196,538 |
| Gridding resolution | 5 m×5 m. |
| Mean points per DEM pixel | 299 |
| Post-processing | |
| Mean stable terrain NMAD before/after co-registration* | 0.32 m a$^{-1}$ / 0.13 m a$^{-1}$ |
| Generated DEMs | 1,907 out of 2,364 stereo-panoramas (81%). |

**Table 2.** Swiss-wide change rates and total changes in specific mass balance (B), thickness, area, volume and mass from 1931–2016. See equations 2–5 for how these are calculated. The uncertainties are the 95% confidence intervals.

| Parameter | Magnitude ± uncertainty |
|---|---|
| Total ca. 1931 area (km$^2$) | 1492±68 |
| Total ca. 1931 volume (km$^3$) | 121.1±6.7 |
| Mass balance rate (m w.e. a$^{-1}$) | –0.52±0.09 |
| Area change rate (km$^2$ a$^{-1}$) | –6.2±0.8 |
| Volume change rate (km$^3$ a$^{-1}$) | –0.73±0.08 |
| Mass change rate (Gt a$^{-1}$) | –0.62±0.14 |
| Total mass balance (m w.e. ) | –43.8±8.0 |
| Total area change (km$^2$) | –531±69 |
| Total volume change (km$^3$) | –62.4±6.9 |
| Total mass change (Gt) | –53±12 |
| Total relative area change | –35.6±6.5% |
| Total relative volume change | –51.5±8.0% |

**Table 3.** The relative importance of each glacier variable in a multivariable regression to explain the variance of glacier mass balance (c.f. Huss, 2012). The total $R^2$ of the multivariable regression using all parameters is 0.46, and the " new $R^2$" column for each parameter is the explained variance when excluding the parameter in question, and "$R^2$ diff(-erence) is" the difference in percent to the total "$R^2$". The coefficient sign shows if the parameter is positively or negatively correlated with glacier mass balance.

| Parameter | New $R^2$ | $R^2$ diff. | Coefficient sign |
|---|---|---|---|
| Median elevation (m a.s.l.) | 0.24 | 48.3% | + |
| Area (km$^2$) | 0.29 | 37.5% | - |
| Lower 10% slope (°) | 0.42 | 8.2% | + |
| Debris cover (%) | 0.45 | 1.4% | - |

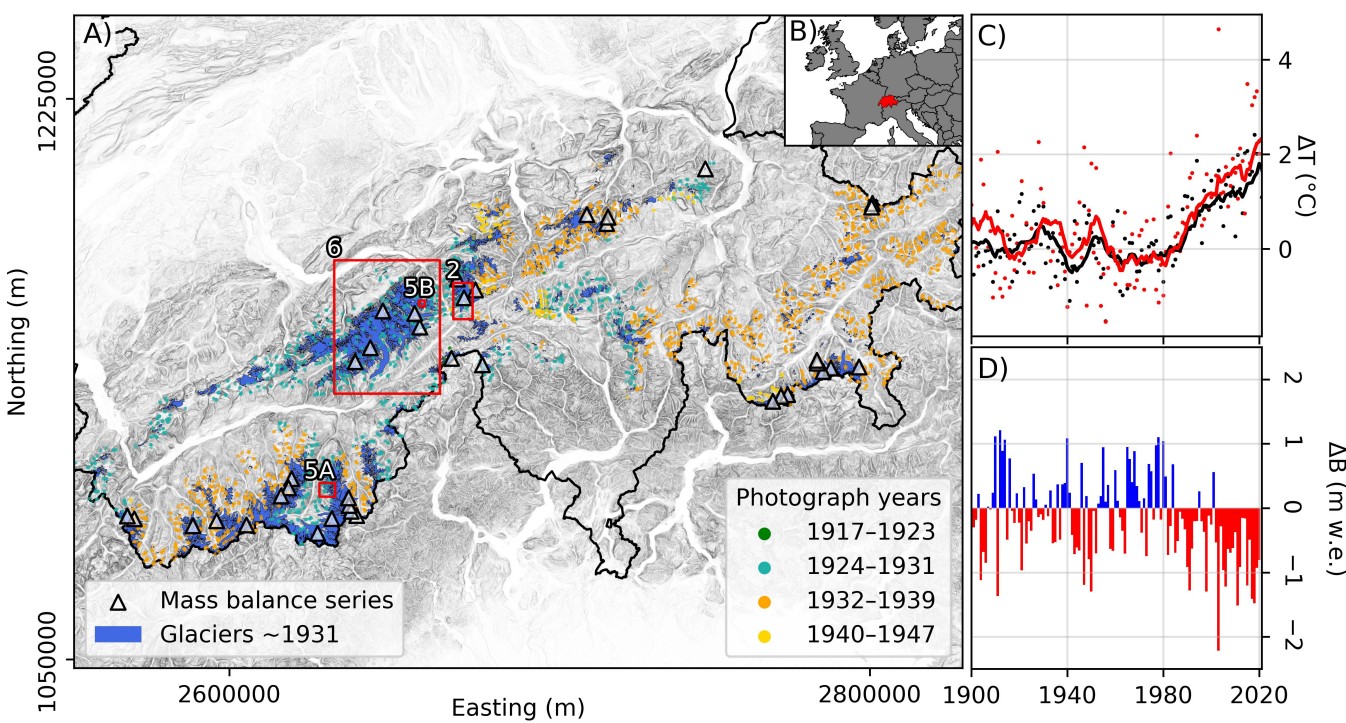

**Figure 1. A)** The distribution of Swiss glaciers around 1931 (blue areas). Country borders are outlined in black. The locations of the TerrA photographs used in the study are shown with coloured dots, whose colour indicates the year of acquisition. The red boxes show the bounds of other figures (figure numbers given in white). **B)** The location of Switzerland (marked in red) in Europe. **C)** Anomalies of mean annual (black) and mean summer (red) air temperature from 1961–1990. Ten year running averages are drawn as lines. The data are from 14 homogenised stations (triangles in **A**, MeteoSwiss, 2021). **D)** Mean anomalies in annual glacier mass balance from 1961–1990. The data are described in Sec. 2.3. Map coordinates are in the Swiss CH1903+ / LV95 (EPSG:2056) coordinate system.

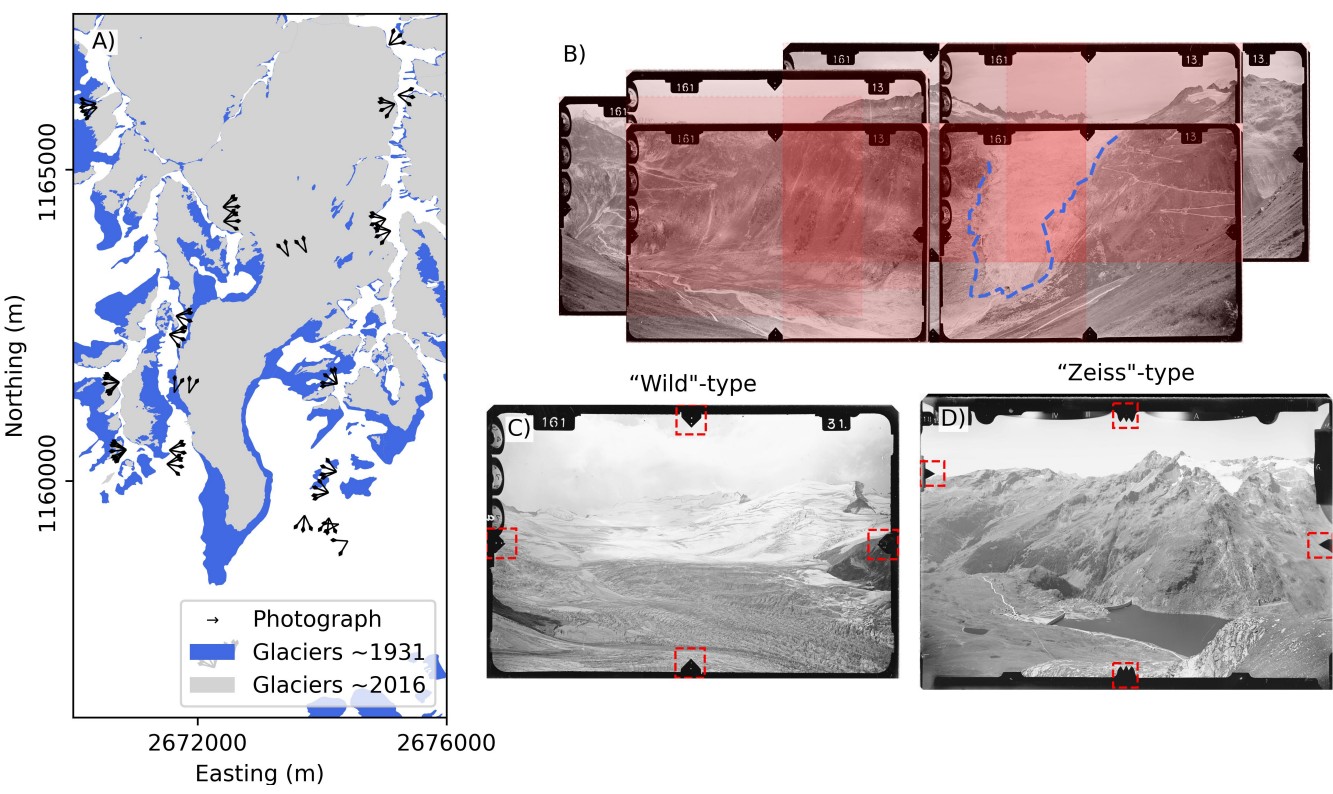

**Figure 2. A)** Imaging locations (black arrows pointing in the imaging direction) together with ca. 1931 (blue) and ∼2016 (grey) glacier outlines for Rhonegletscher. **B)** Panorama example overlooking Rhonegletscher (outlined in dashed blue). The opaque red boxes show the coverage of each image. The darkest red indicates five images overlapping. **C+D)** Examples of a "Wild"- (**C**) and a "Zeiss"-type (**D**) photograph, and associated fiducial marks (framed in red).

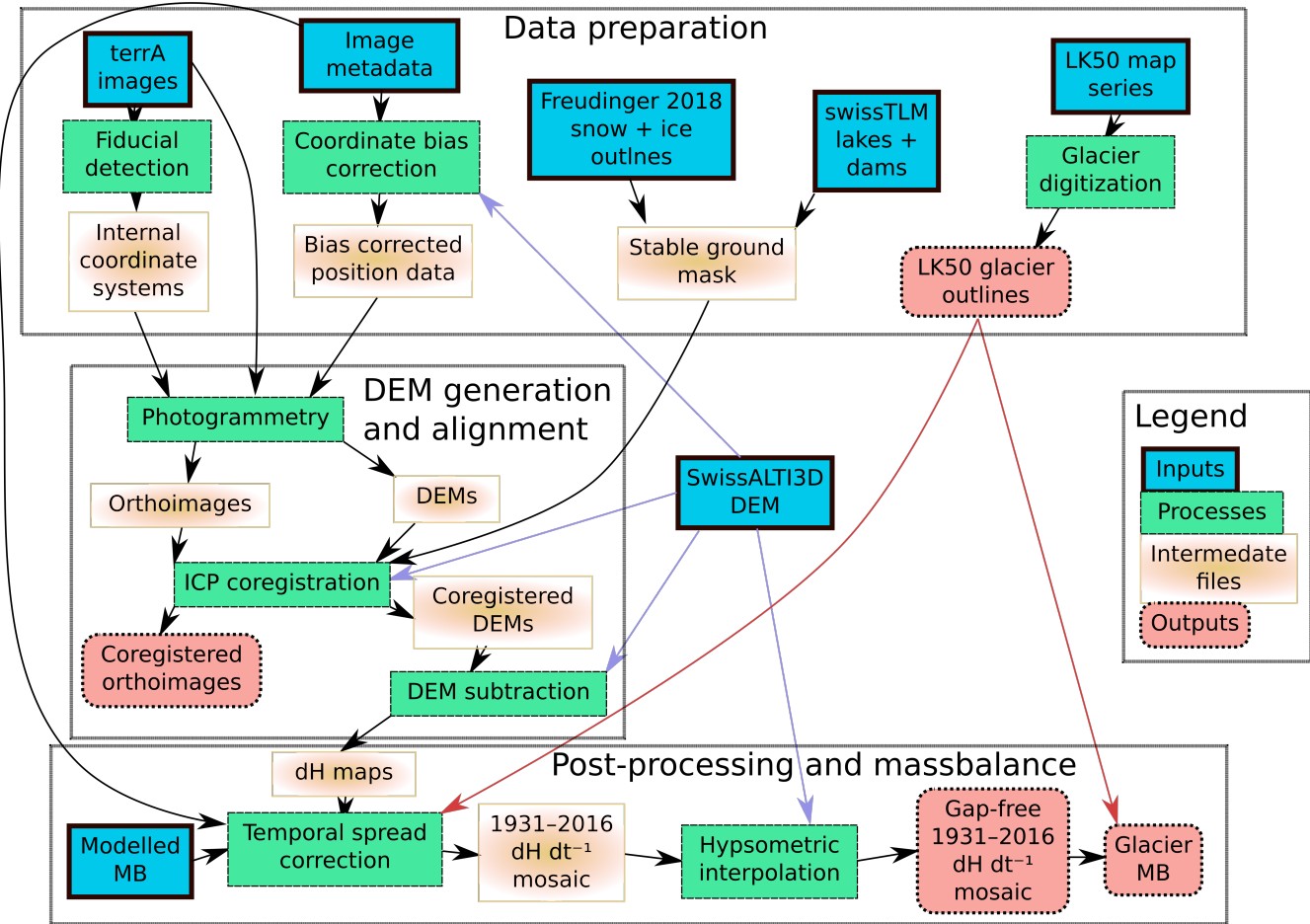

**Figure 3.** Flowchart synthesising the data processing. Inputs refer to the external data that were used, processes are different steps in the processing chain, and outputs are files of particular interest for this and future studies. All intermediate and output files are available for further use.

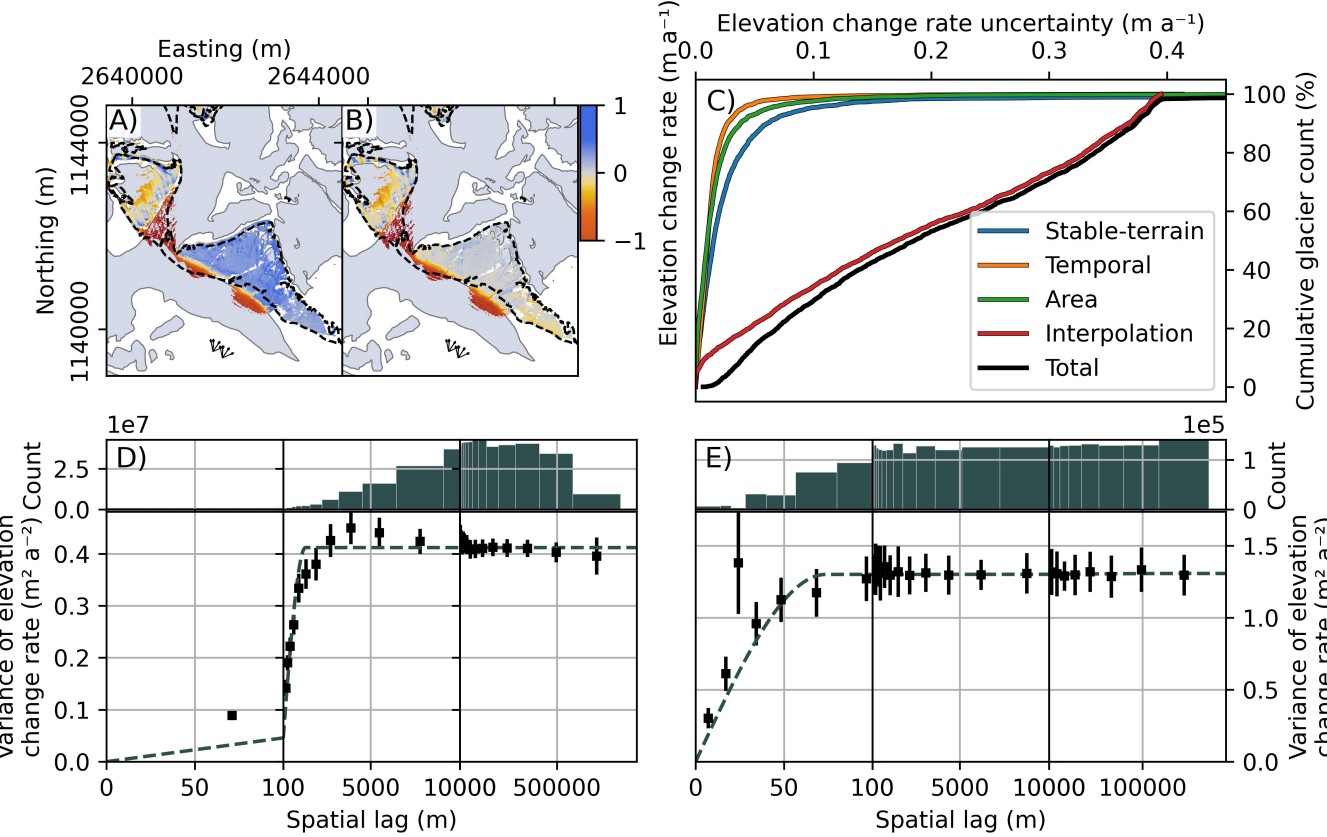

**Figure 4.** Error figure ensemble. **A+B)** Differences of an individual DEM before (**A**) and after (**B**) coregistration. The arrows indicate the locations and directions of the images used for generating the ca. 1931 DEM. The dashed lines indicate the theoretical viewshed of the stereo-pair (pre-calculated by swisstopo). **C)** Cumulative histograms showing the magnitude of the different components for the elevation change uncertainty of each individual glacier. **D+E)** variograms of interpolated (**D**) and stable-terrain (**E**) elevation difference uncertainties, showing the variance of all pairs of pixels at a given distance (or lag) as a function of that lag. The individual markers show the empirically derived variance, their error bars show their 95% confidence intervals, and the dashed line shows the variogram model (sum of two spherical models). The histograms above the variograms show the pairwise sample count for each empirically derived variance marker.

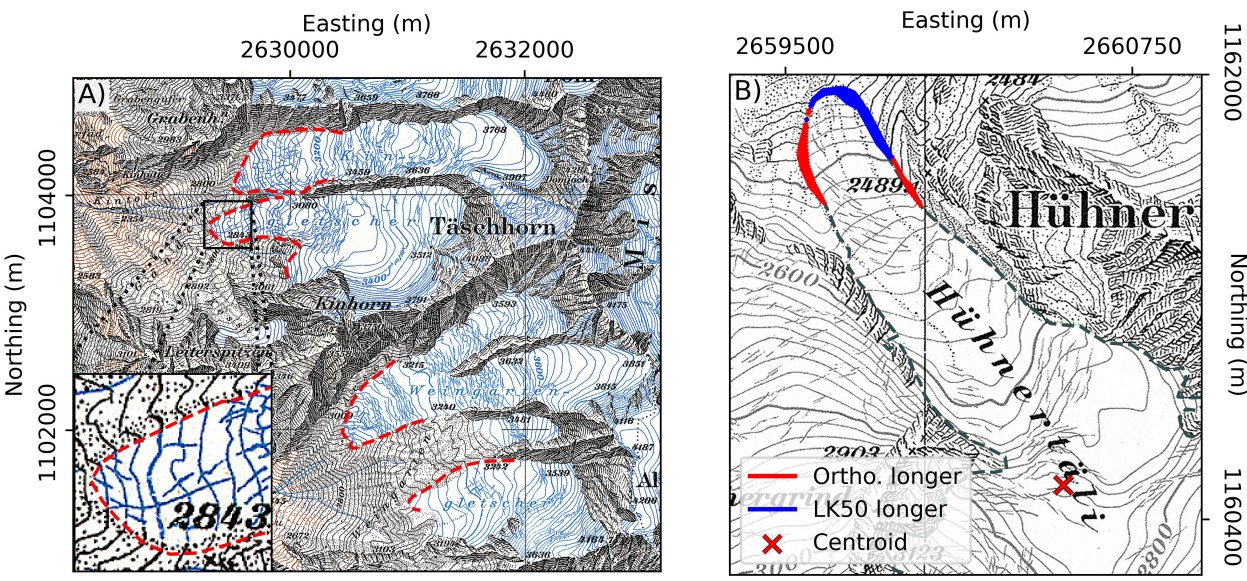

**Figure 5. A)** Glacier outlines drawn from orthoimages generated in this study (dashed red), superimposed on a sheet of the LK50 map series.
**B)** Comparison between glacier outlines derived by digitising the LK50 map series and orthoimages. Red shading indicates that the outline drawn from the orthoimage is farther away from the centroid than the outlines derived from the LK50 map series. Blue shading indicates that they are closer.

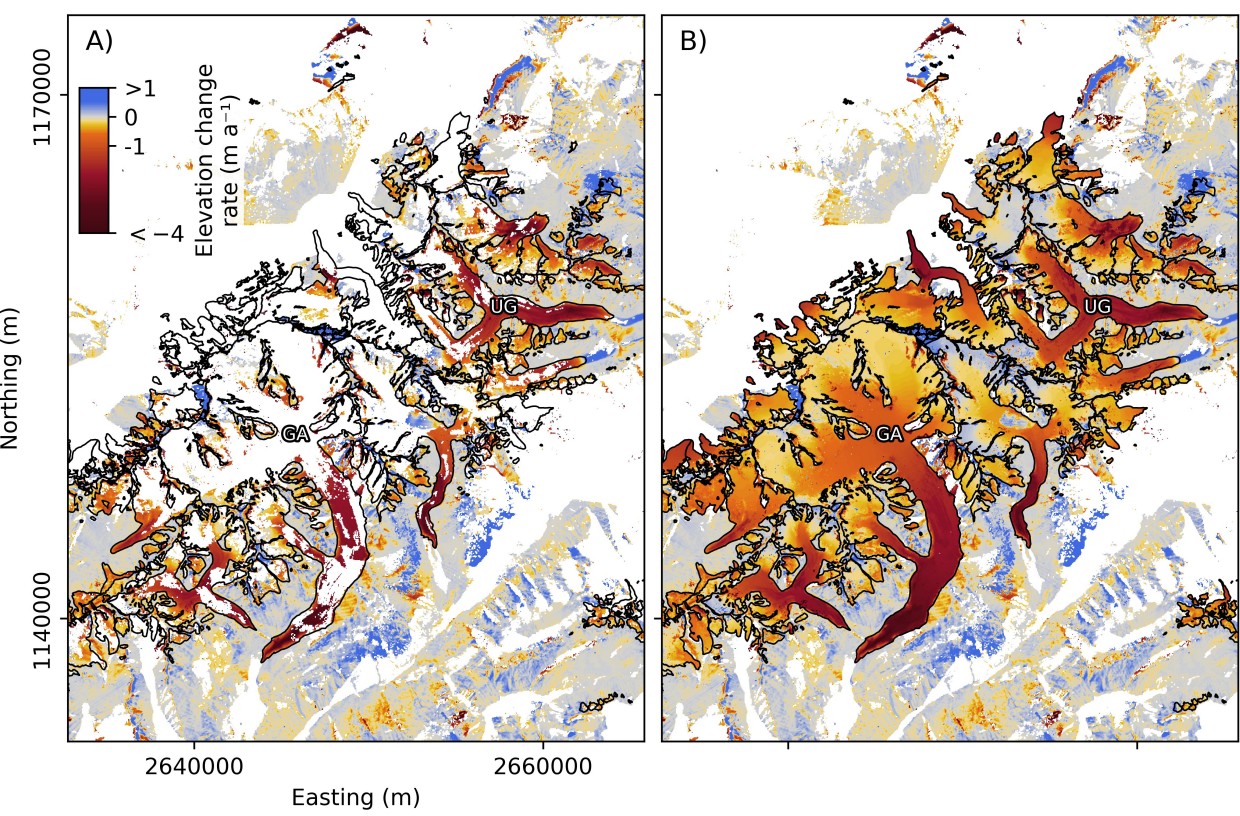

**Figure 6.** Maps of average ice thickness change rates (dH dt$^{-1}$) over the period 1931–2016 before (**A**) and after (**B**) interpolation. The area shows Grosser Aletschgletscher (GA), Unteraargletscher (UG) and other neighbouring glaciers. Glacier outlines from ca. 1931 are drawn in black.

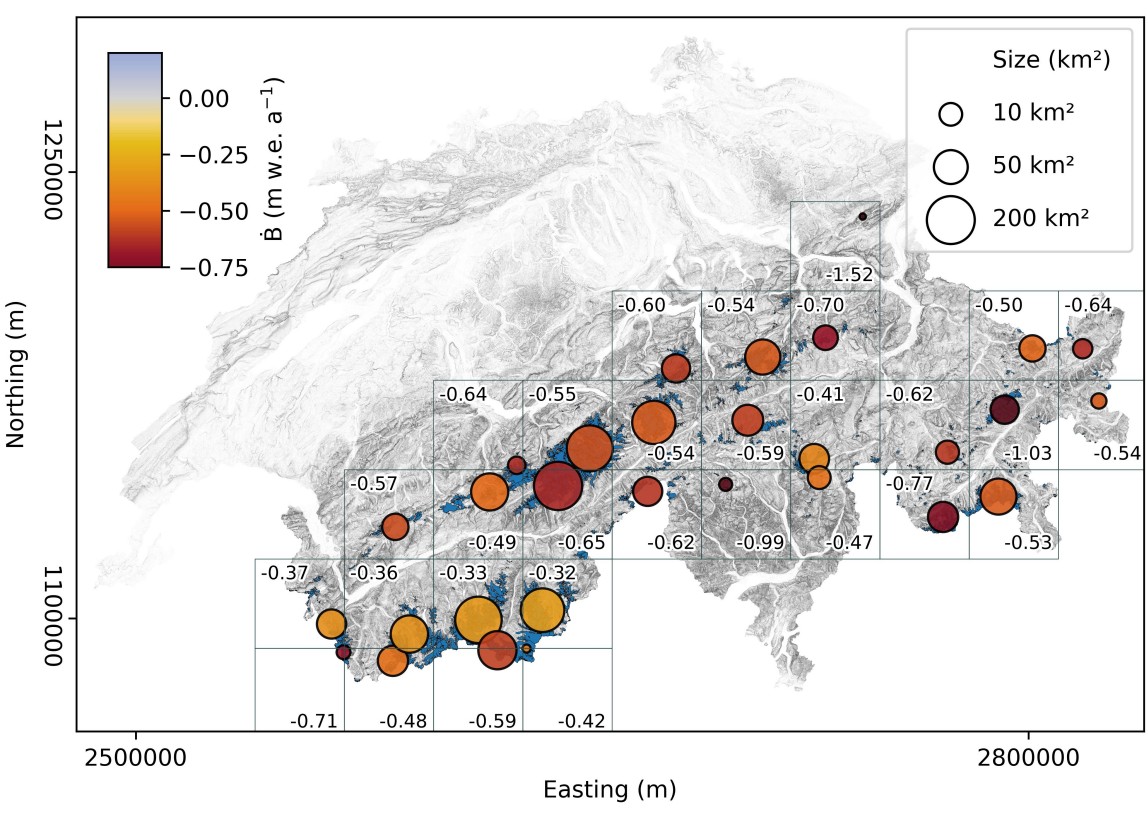

**Figure 7.** Area-weighted geodetic mass balance rates (colours and numbers) for $30\,\text{km} \times 30\,\text{km}$ tiles (grey squares). The location of each circle is the average glacier centroid weighted by area in the tile, and their sizes indicate the glacier area in ca. 1931.

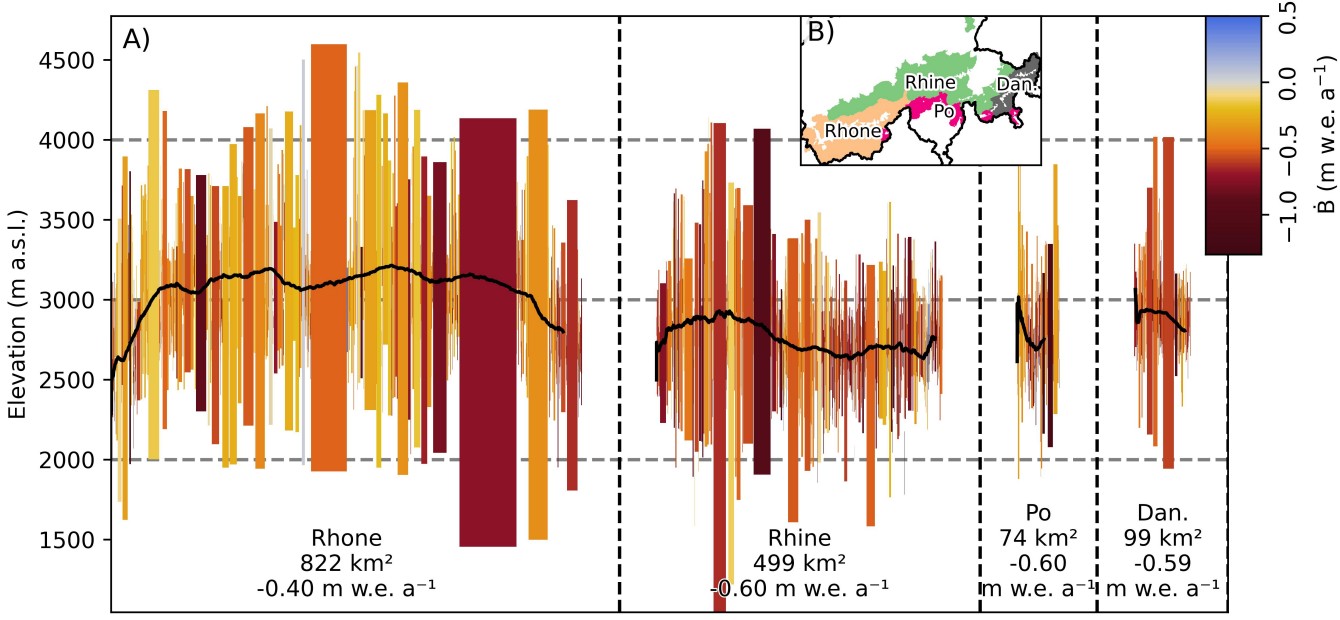

**Figure 8.** West-to-east transect of glaciers in Switzerland (**A**), grouped by the four main drainage basins (**B**). Each bar represents a glacier; the width indicates the ca. 1931 glacier area, the colour shows the average geodetic mass balance from 1931–2016, and the lower and upper bounds show the glacier's modern elevation range. The near-horizontal black lines are moving averages of the median glacier elevations. For each drainage basin, the total area as of ca. 1931 is given, together with the average geodetic mass balance weighted by glacier area.

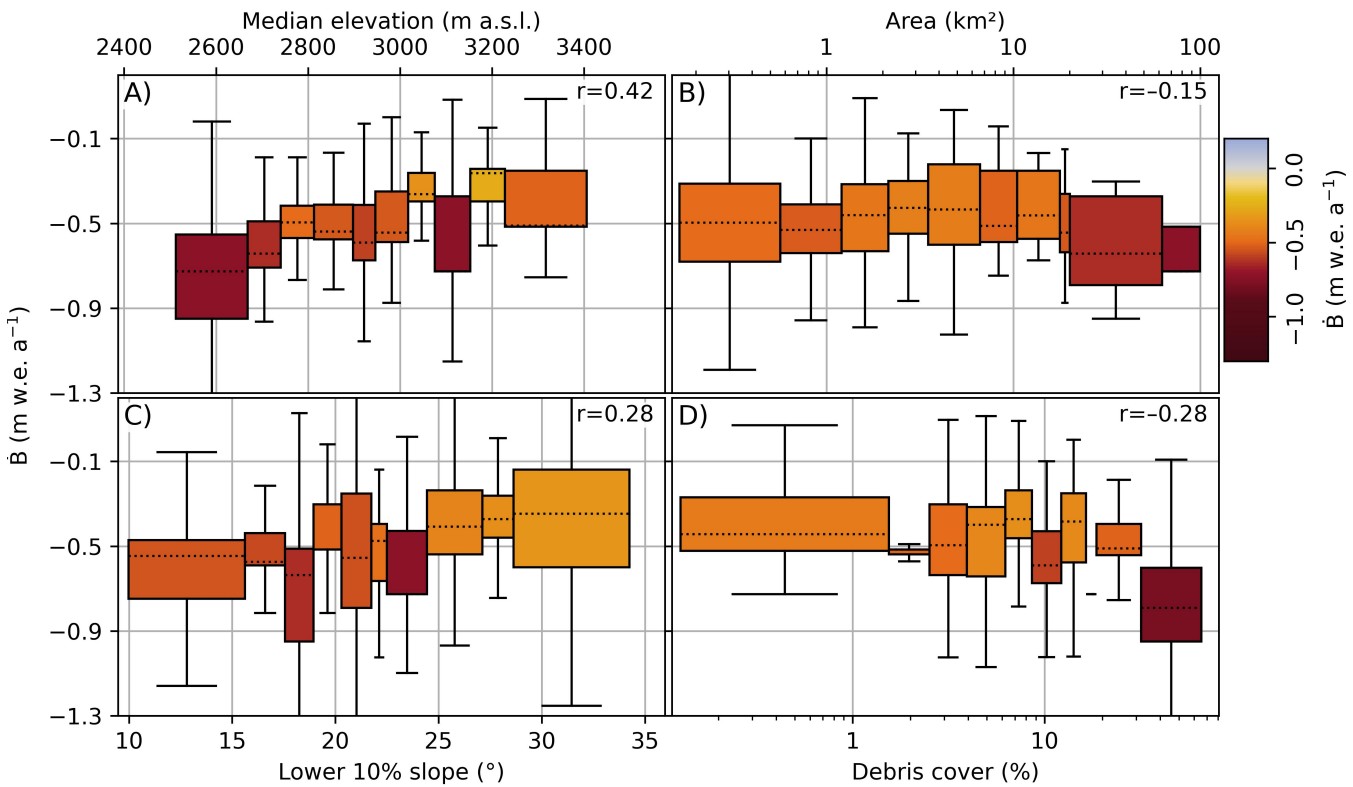

**Figure 9.** Area-weighted 10th percentile (0–10%, 10–20%, etc...) boxplots showing the relations between 1931–2016 average glacier mass change rates (vertical axes and colours) and different morphological parameters (horizontal axes). The whiskers extend from the box by 1.5× the inter-quartile range (IQR). The stated Pearson correlation coefficients (*r*) refer to the unbinned variables. **A)** Correlation with median glacier elevation, indicating that lower elevations show larger mass loss rates. **B)** Correlation with glacier area, showing no clear relation. **C)** Correlation with the slope of the lowermost 10% of the glacier surface, possibly showing higher mass change rates for glaciers with flatter termini. **D)** Correlation with fractional debris cover in 2016 (taken from Linsbauer et al., 2021), indicating that higher mass change rates may be found for glaciers with high debris-coverage percentage.

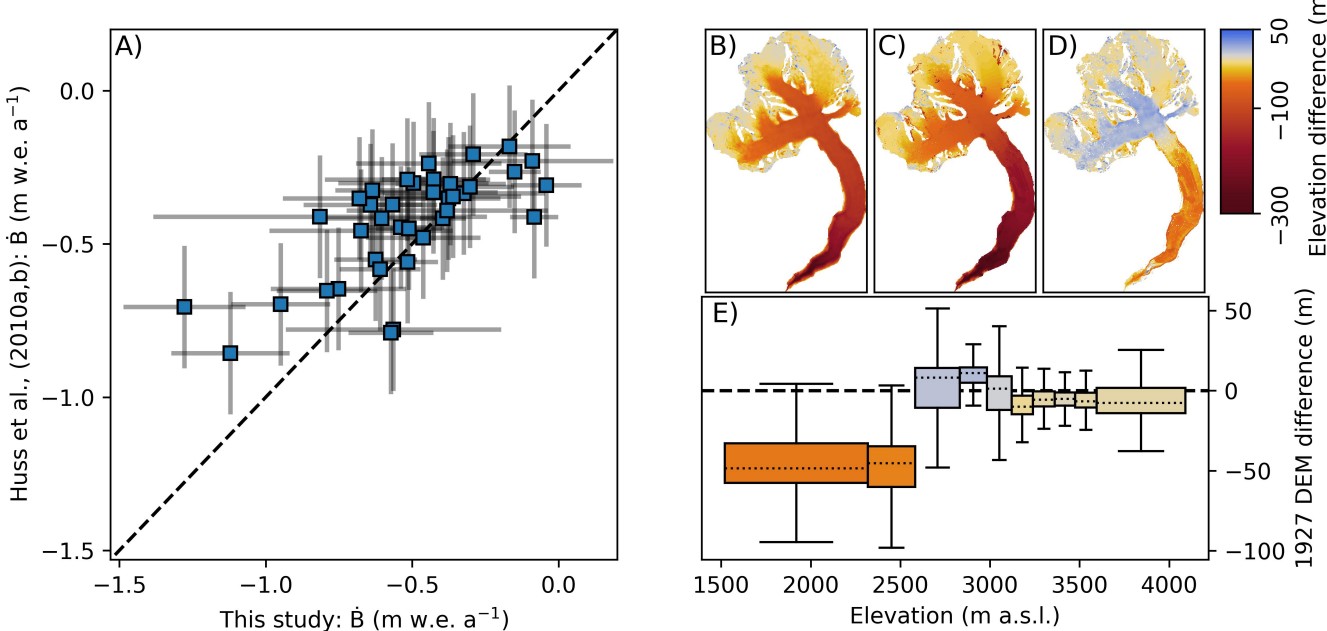

**Figure 10. A)** Comparisons in the same time-intervals of 39 glacier-wide mass balances inferred in this study with mass balance derived from modelling, constrained by geodetic surveys at decadal intervals (blue squares; Huss et al., 2010a, c). The Pearson correlation coefficient between the points is 0.71. The horizontal uncertainty bars are the 95% confidence intervals of this study, the vertical uncertainties are fixed uncertainties of 0.2 m w.e. a$^{-1}$ (see Sec. 5.1), and the dashed line is a 1:1 line. **B)** Elevation difference between the 1927 topographic map digitised and used by Bauder et al. (2007), and the SwissALTI3D DEM in 2017. **C)** Elevation difference between the DEM mosaic generated in this study and the SwissALTI3D DEM in 2017. **D)** Difference between the 1927 topographic map and the DEM mosaic from this study (map – this study). **E)** Differences between the two 1927 DEMs (c.f. **D**) binned for every 10th percentile of elevation. A general agreement is seen on the upper part of the glacier (above ∼2700 m a.s.l.), but the tongue is portrayed as smaller in the topographic map.