# Peer review of "Halving of Swiss glacier volume since 1931 observed from terrestrial image photogrammetry"

_The Cryosphere, 2022_

## Referee Comment (RC3)

**Review for "Halving of Swiss glacier volume since 1931 observed from terrestrial image photogrammetry"**
**Authors: Erik S. Mannerfelt, Amaury Dehecq, Romain Hugonnet, Elias Hodel, Mattias Huss, Andrea Bauder, Daniel Farinotti**

**General Comments:**

Study uses tens of thousands of terrestrial photographs collected from ~85% of the glaciers located in Switzerland to generate historical surface elevations and determine ~85-year mass balance change. The amount of work performed in this study is extraordinary and the results are very exciting. The methodology used by the authors for elevation derivation, calculating mass balance, and estimating error is sound. The majority of my comments in the manuscript are to do with seeking a bit more clarification in the text.

"Historical" and "Historic" are used throughout. "Historic" is subjective on the authors' part, so the more appropriate adjective is "historical".

The word "taken" in referring to image acquisition is colloquial. It is better to use "acquire" or "collect" instead.

In Section 2.3, because glacier outlines are being derived from historical topographic maps and datums/ellipsoids evolved constantly throughout the timeline of this study, it is necessary to mention whether digitized versions of the Siegfried maps and the LK50 maps are currently available in their original horizontal projections, or if transformations were preformed so they align with the more current datasets like the swissALTI3D DEM. Admittedly, I am not that familiar with the historical Swiss topographic series, but I am assuming that a different datum was used in its assembly than what is in practice today. This is certainly the case with the American 7.5 minute series topo maps, where XYZ values derived using NAVD29 have a horizontal shift of ~400-500 m for regions of greater topographic terrain variability. For the sake of transparency of potential error sources, it is necessary to at least mention whether all datasets were transformed to the same map projection.

With several recent studies using datasets comprised of multi-temporal acquisitions (Rignot et. al., (2021); Haubner et. al., 2018; De Rydt et. al., 2021; etc.) as discrete timestamps, I very much appreciate the authors' efforts to standardize the historical elevations' timeline. I wish more studies were as arduous and feel this paper is an example of "best practices" that I hope future multi-temporal glaciological research would benefit by following. The one addition that I request be included with the temporal error assessment is to show the absolute differences with a sample of dh results from specific timestamps to the standardized timeline.

The state of the manuscript is that of a methodology paper. If that is the intent of the authors, then ignore this next part. However, the impact of this paper could be significantly improved if in the discussion section you were to include the broader impacts of your mass balance results with topics like future predictions of Swiss glacier volumes, future melt-water flux to stream discharge rates, anthropomorphic impacts from 1916-2016, etc. You've achieved an amazing

temporal scale for mass balance changes over such a large spatial extent that it would be beneficial to the glaciological community if you could include its relevance in a broader scope.

**Specific Line Comments:**

Line 8: I think you mean "conducive" rather than "conductive"

Line 20: "understand" is odd in this context. "resolve" would be a more appropriate word.

Line 33: Does Switzerland refer to the country or its glaciers?

Line 40: does "photographed position" refer to the camera's position or the image's full extent?

Line 44: using the term "re-process" is confusing. I am assuming that you are referring to the processing that was done with these images to generate the original topo maps in which case it would be better to say "digitally process" instead of "re-process".

Line 53: Replace "houses" with "contains"

Line 56: Delete "of"

Line 69: Replace "will be" with "is"

Line 76: Replace "5" with "five" for continuity

Line 77: It is unclear exactly what "frame appearances" means. Maybe "frame geometry" would be better?

Line 126: Can you go into more details about the similarity transformation and how it is performed?

Line 129: Is "marks" referring to fiducial marks?

Line 132: "Manual data" is awkward; is the model that is being referenced the same as the previous sentence?

Line 137: "manually mark" is unclear. Do you mean "identify"?

Line 144: Is the metadata digital or analog? Where do the image positions come from?

Line 150-151: Verb tense is in the past. Also, the difference value being reported, is that a vertical difference? Can you also provide the horizontal difference?

Line 152: Did you perform the map digitization? If not, then the heading should reflect generating glacier outlines instead.

Lines 153-163: Please include somewhere in this section what the average glacier area is to provide scaled-context for researchers primarily doing work in Greenland and Antarctica.

Line 153: "digitizing the scanned and georeferenced LK50 map series." It is unclear here whether the maps are being digitized or the glacier outlines.

Lines 154-155: The verbs in this sentence are written in the past tense. Also, use "sporadic" or "inconsistent" rather than "spotty"

Lines 161-161: The verb in this sentence are written in the past tense.

Lines 165-166: Suggested rewrite for "version 1.6.5 by dividing the images in subsets, and processing each subset individually (Table 1)." with "version 1.6.5 in separate subsets (Table 1)."

Line 166: Replace "one" with "each", "taken" with "acquired", "certain" with "specific", and "at a certain" with "over a single".

Line 167: Should "groups" be "subsets"? If not, then you need to define what a "group" is.

Lines 167-168: Replace "subset, as the used cameras might vary in their distortions parameters, depending on their construction and in which year they were used;" with "subset because of varying distortion parameters for each camera;"

169-170: Do you have evidence to support this statement? It really should have a citation or maybe documentation from historical metadata.

Line 171: Are stereo-panoramas one per subset? Are these the camera models generated in the previous paragraph? If so, then say something like "the computed camera-models" so there is no question that potentially new camera-models have been introduced.

Line 172: It is unclear what "others" is specifically referring to. Is it other "panoramas" and/or is it other image subsets? This also applies to "them" on Line 174.

Line 177: In referring to the co-registration, you need to define what is being co-registered in regards to stable terrain or over entire generated data coverage.

Lines 180-187: In determining the conversion of DEM grids from 3D point clouds, how did you decide on the cell size? What did you determine for a cell size? Was the entirety of each point cloud utilized in DEM conversion? Studies using oblique aerial imagery have found that there is a distance from point of image acquisition that the generated elevations are no longer reliable (Girod et. al., 2018); was this phenomenon a factor with the terrestrial imagery? If so, was this consistent across all historical acquisitions?

Line 180: Is the stereo-panorama alignment being applied to all panoramas generated from the ~21,000 images or to just panoramas from a single subset?

Lines 182-183: It is untrue that details for the confidence parameters are not available. White papers written by Agisoft Metashape are readily available online that describe how the confidence parameter is determined based on the contributing depth maps. Simply reporting a value of "2" without any supporting evidence to validate its use systematically appropriate is imprecise—please amend this.

Lines 183-184: It is unclear what relevance draping the orthoimages over the DEMs has for the processing and I recommend deleting "and after orthoimages are draped on the DEMs," from the sentence.

Line 186: Alter "position and rotation" with "positional and rotational errors"

Line 187: Include "historical" before "DEMs".

Line 189: I suggest calling the "correction" an "adjustment" or "standardizing" because time itself is not in error.

Lines 190-192: The parameters described in the first two sentences have already been explained previously in the manuscript. You are not technically correcting the temporal scale, but instead standardizing it to specific years. It would be better in this paragraph to explain why the temporal scale requires standardizing, the specific timeline of standardization for each dataset, and the potential errors that may arise when rates are not based on the absolute temporal scale.

Line 193: Alter "correct" with "adjust"

Line 195: Delete "anomalies in" because you are using the full extent of mass balance calculations and not just anomalies.

Line 198: It is unclear if "respective region" refers to each of the four glaciers separately or all of them lumped together.

Line 199: It is unclear with the "factor" is.

Line 200: Alter "corrected" with "standardized"

Line 205: Alter "corrected" with "standardized"

Line 207: State whether the percentage of missing data is the result of image coverage extent or processing, or both.

Line 208: Delete "the whole"

Line 209: Can you quantify "significant trend"? What is the trend in reference to? Verb use is in past tense.

Line 212: Referencing "xdem" needs a citation or a reference that it is a python module.

Line 214: What is the "threshold" in reference to? This needs more details and if the threshold is based on other research, a citation as well.

Line 215: Delete "the"

Line 216: Are full data gaps replaced with the single value or is there a maximum distance where new values are not assigned when using this interpolation technique? Please include either scenario in the text.

Line 218: Recommend altering "each entry of the SGI2016 individually." with "each SGI2016 outline."

Line 221: Alter "historic" with "historical"

Line 236: What are the "elevation measurements"? Are these the generated historical elevations? If so, then say "stochastic processed elevation errors" instead because "elevation measurements" sounds like altitudes collected *in situ*. Otherwise, state where these measurement come from.

Line 237: Alter "corrected" with "standardized"

Line 239: Alter "presented through" with "within"

Line 240: Suggested rewrite of "from the scale of pixels to that of glacier, and that of glaciers to regional scales," with, "from scales of pixel to glacier and glacier to region,"

Line 250: For the stable terrain, do you include all surface slopes as well?

Lines 257 and 258: Phrases "might be affected" and "might cancel out" are too uncertain and do not support your argument. Either provide a reference to past research who has identified similar results, or a statistical assessment to be able to assert that the quantity of DEMs in this study is indeed the reason DEM differencing is unaffected by terrain parameters like slope. Also, "these" on Line 258 referring to the attribute-dependent errors?

Lines 270-272: What about the maps' error? There absolutely was error in the *in situ* triangulation estimates and original photogrammetric work—are there no estimates of the actual maps' accuracy?

Line 272: Are there no references as to the years of data collection by map? Do the maps not contain this information? Were any reports written with the topo maps? US and UK topo maps contain temporal information and often had supplemental reports written to validate methods used when compiling the maps.

Line 273: It is unclear what "degree" is specifically referring to.

298: "Polluted" is an odd word choice. I recommend using something like "biased" instead.

Line 301: It's annoyingly pedantic, but "small scale" is actually referencing a very large extent. For continuity with the rest of the text, it would be better to reference either "pixel-scale" or "glacier-scale".

Lines 302-303: "largely independent among glaciers of a given region." Does this mean the errors are independent by glacier, or by region?

Line 305: Recommend altering "correction" to either "adjustment" or "standardized"

Line 307: Is this uncertainty based on mass balance derived from remote sensing methods? If so, that should be stated.

Line 321: The minus symbol is unnecessary because you say there is a "loss".

Line 323: For continuity in the text, please also provide the 1931 glacier volume.

Line 341: Are these "glaciers" also Swiss? Please provide a regional reference to the glaciers.

Line 351-353: This sentence is difficult to understand, particularly the portion on Line 353.

Line 354: What specifically are the "components" referring to?

Line 356: Suggest altering "when considering the entire period of interest." with "for the study's timeline."

Line 356: Suggest altering "The comparison to results of the present study" with "Comparing those results to this study's".

Line 358: Replace "smaller" and "study" with "less" and "estimates" respectively.

Line 359: What year is the map data from? Is it the same as the imagery? Was the map data and the derived historical DEM data referenced in the same vertical datum? If they weren't, then what transformation was used with the map data?

Line 405: Alter "improving" to "improve"; is "description" the word you meant here? If so, it is unclear what you are specifically referring to.

Line 432: Include commas after "approach" and "ago"; verb tense in this sentence is past.

Line 446: "high potential" is awkward.

**References:**

De Rydt, Jan, Ronja Reese, Fernando S. Paolo, and G. Hilmar Gudmundsson. "Drivers of Pine Island Glacier speed-up between 1996 and 2016." *The Cryosphere* 15, no. 1 (2021): 113-132.

Girod, Luc, Niels Ivar Nielsen, Frédérique Couderette, Christopher Nuth, and Andreas Kääb. "Precise DEM extraction from Svalbard using 1936 high oblique imagery." *Geoscientific Instrumentation, Methods and Data Systems* 7, no. 4 (2018): 277-288.

Haubner, Konstanze, Jason Box, Nicole Schlegel, Eric Y. Larour, Mathieu Morlighem, Anne Solgaard, Kristian K. Kjeldsen et al. "Simulating ice thickness and velocity evolution of Upernavik Isstrøm 1849-2017 with ISSM." In *AGU Fall Meeting Abstracts*, vol. 2017, pp. C11B-0911. 2017.

Rignot, Eric, Lu An, Nolwenn Chauche, Mathieu Morlighem, Seongsu Jeong, Michael Wood, Jeremie Mouginot et al. "Retreat of Humboldt Gletscher, north Greenland, driven by undercutting from a warmer ocean." *Geophysical research letters* 48, no. 6 (2021): e2020GL091342.

---

## Author Comment (AC1)

**Contents**

| 1        | Overview                                                                                     | 1                  |
|----------|----------------------------------------------------------------------------------------------|--------------------|
| 2 | Editorial comments                                                                           | 1                  |
| 3        | Community comments 1 (CC1)                                                                   | 2           |
| 4        | Referee comments 1 (RC1)                                                                     | 3                  |
| 5        | Referee comments 2 (RC2)5.1 Specific comments5.2 Technical comments/corrections (line edits) | 4
5
7 |
| 6        | Referee comments 3 (RC3)   6.1 Specific Line Comments:                                       | 8
9      |

**Colourcode**

A solved comment A rejected comment (with an explanation)

**A comment reply**

**1 Overview**

We would like to thank the reviewers for their constructive comments that have really helped improve the quality of the manuscript and the editor for giving us the opportunity to address these comments in a revised version. We have made several significant changes to the text summarised here:

- we have updated figures 1 and 4 following RC2's suggestions
- we have clarified the methodology, in particular regarding the stereo processing (following questions from RC2 and RC3) and the gap filling (following CC1)
- we have largely expanded the discussions in particular to 1) better compare our regional estimate with previous studies (section 5.3), 2) expand on the analysis of the variables explaining the spatial variability (section 5.3) and 3) discuss the complementarity of the in situ, modelling and geodetic approaches to resolve a homogeneous regional glacier mass balance (section 5.4).

We have also made many smaller edits, explained in details below.

Erik Mannerfelt and co-authors.

**2 Editorial comments**

Title: The title should better reflect the period addressed and clarify that 1931 is a mean date. I suggest to change to (or similar) "Halving of Swiss glacier volume during  $\sim$ 1931 and 2016 observed from terrestrial image photogrammetry"

While we agree with the editor that 1931 is a median date for the reconstructed DEMs, our volume change estimate is corrected to *exactly* match the 1931-2016 time period. We therefore believe that it would reduce the clarity if we were to update the title and abstract as suggested. The point was not raised by any of the reviewers either.

L2: Similarly you should make already here clear that 1931 is a mean date

As mentioned with the response to the title suggestion, our reconstruction is in fact between 1931 and 2016. After the suggestion by reviewer RC1, it is now clear from the following line that 1931 is the *median* date for the photographs: "Our analysis relies on a terrestrial image archive known as *TerrA*, which covers about 86%

of the Swiss glacierised area with 21,703 images acquired during the period 1916-1947 (with a median date of 1931)" (L4).

L15: Maurer et al. 2019 is a great study but the major improvement is that the authors present mass balance since the 1970s not since about 2000. Moreover, I suggest to include at least one paper investigating a large area outside of HMA (e.g. there are two which cover the whole Andes.)

Replaced the Maurer et al. (2019) study with Braun et al. (2019) that focuses on South America.

L19ff: I fully agree with this statement about the inclusion of more historical data. Best would be, however, not only to present one long period but several subperiods covering a long time. This should be better mentioned in the intro and also better considered in the discussion section. There are several related studies existing (some of which are mentioned later in this section). But this fact should also be mentioned here and I think it would make sense to consider also Mölg et al. (2019), TC who present a detailed time series for one glacier in the Swiss Alps based on different historical data and Bhattacharya et al. (2021), Nat. Comm. who present a time series since the 1960s for different parts in HMA based on historical satellite data and similar one for other regions in the Himalaya or elsewhere. But I leave this to the authors as I am co-authoring some of these studies.

We agree with the editor that not only longer-term but also more frequent estimates of glacier changes are needed. We therefore slightly reworded that part of the introduction ("To resolve these discrepancies, longer and more frequent observations of glaciers changes are essential, and historical data (i.e. film based images) should be leveraged to better constrain the response of glaciers to a changing climate.") and largely expanded on the limitation of our "snapshot" estimate in the discussion section 5.4. In particular, we added a sentence to acknowledge studies in Switzerland that provide multi-temporal elevation observations: "Note that by leveraging different data sets, it is possible to obtain more frequent observations, spaced from a few years to a few decades apart, to help constrain the multi-decadal variability (Rastner et al., 2016; Mölg et al., 2019." (L489).

L53: Please cite also the original source of the data which was used in RGI6.0 and include the information from the most recent glacier inventory of the Alps (I think it is Paul et al. 2020, ESSD).

Replaced the RGI6 citation with Paul et al., (2020)

L. 205: Use past tense "generated"

We kept the present active voice "generate" for consistency with the rest of the Methods.

**3 Community comments 1 (CC1)**

Very nice and interesting work. I only have some questions regarding the interpolation.

Thank you for your positive comment and your constructive questions.

L210 ff: You talk about "regional hypsometric approach" and refer to McNabb et al. (2019). However, McNabb et al. (2019) use the terms "local" and "global" approaches. I guess you mean the local approach, i.e. an interpolation for each glacier individually. Did I get it right? You applied also scaling of the elevation range and elevation changes per glacier. How did you define the size of the elevation bins used for the interpolation and what's the advantage of the scaling?

In our study, the "regional hypsometric interpolation" is equivalent to the "global hyspometric interpolation" presented in McNabb et al. (2019). We added a clarification on the terminology in the text "We use a modified version of the regional (also referred to as "global") hypsometric approach". We believe the term "regional" is more appropriate since it would not be applicable at the global (i.e. worldwide) scale. We used a mix of a local (when sufficient observations exist) and regional approach (in the other cases). Further details can be found in the Methods. We have added that the bins are 5th percentiles of elevation and elevation change. The scaling with normalized elevation and normalized elevation change helps minimize regional inconsistencies during interpolation that are found with the global hypsometric method (e.g. Dussaillant et al., 2019). This normalized hypsometric relationship mirrors that used in modelling studies, e.g. Huss et al. (2010b).

In L215 you talk about glaciers with >20% voids. Did you apply here the "global" hypsometric approach, as defined by McNabb et al. (2019).

Yes, and we hope that this is clearer now that we have added in the text that "global" and "regional" are synonymous (L223)

L303: How did you propagate the interpolation uncertainty from pixel to glacier and regional scales? We used Equation 6 to estimate a number of independent samples in interpolated errors, later propagated through Equation 9. We have streamlined the associated statement in the manuscript.

Kind regards

Thorsten Seehaus

**4 Referee comments 1 (RC1)**

This is an important and valuable contribution to the analysis of 20th century glacier changes prior to the advent of satellite imagery, and the results are significant. I have only a few minor mostly grammatical suggestions as below:

We thank referee RC1 for the positive feedback and the wording suggestions.

Abstract: line 4: '1931 on average' .. this implies the number of photos are distributed evenly around this date, but this is not demonstrated. It is perhaps safer to describe 1931 as the median year as they have done subsequently.

We changed to say "with a median date of 1931"

Line 26 and 396 'Whilst' : this word is rarely used in normal English, and never in American English. The authors would be better to replace this with the more standard 'while'. We changed to "while" in both occurrences.

Line 55-56: which is of  $\rightarrow$  which are We applied this change

99: cartographic mapping ... simply replace with mapping (cartographic is redundant) We applied this change

103: dozens of  $\rightarrow$  dozen We applied this change

115: delete 'being' We applied this change

124 delete 'used' We applied this change

129: allows  $\rightarrow$  allow We applied this change

215: delete 'the' We applied this change

325: are was of  $\rightarrow$  area was We applied this change

326: delete 'of' We applied this change

 $340 \text{ as} \rightarrow \text{than}$ We applied this change

428: delete 'compared to 2016' ... and change: a third's reduction in areal cover to ... reduction in areal cover by a third

We applied this change

Figures 1,2,4,5,6,10: coordinates are given but not explained. The reader can only assume they are a Swiss coordinate system (LV95 ?), but this should be noted in at least Figure 1.

Fig. 1 now states which coordinate system the coordinates are in (CH1903+/LV95).

**5 Referee comments 2 (RC2)**

Mannerfelt et al. use 21,703 historical terrestrial photographs, acquired during the period 1916-1947, to reconstruct the 3D geometry of 45% of Switzerland's glaciated area nearly a century ago. This paper is an absolute tour de force. I was impressed by not only the sheer scale of the analysis (successfully generating 113 different Agisoft Metashape models, each with an average of 192 historical photographs), but also the level of care and detail in the error propagation and uncertainty analysis. This project provides an impressive and carefully curated dataset that will prove useful to the glaciology community for better understanding the drivers of glacier mass loss. I have two general suggestions, followed by a handful of more specific comments, that I hope can help Mannerfelt et al. improve the presentation of this exciting manuscript.

We thank referee RC2 for his positive comments on our work and particularly his time to help us improve the manuscript presentation. In the following, we address their comments.

First, I encourage Mannerfelt et al. to dedicate a paragraph or two in the discussion to explore the implications of their new mass balance constraints. What have we learned from this new dataset? For example, do the authors have ideas about what controls the regional mass balance patterns shown in Fig. 7? What fraction of that mass balance variability can be explained by temperature/elevation, precipitation, glacier slope, debris cover, etc.? How might these climatic and geometric controls on glacier mass balance strengthen or weaken in the next century?

We thank the reviewer for the relevant suggestion to go further in depth on the drivers of mass balance variability. Two new subsections of discussion have been added (Sec. 5.3 and 5.4) where we further compare to previous studies and try to explain the differences we observe. We also added an analysis to calculate the explained variance of the variables shown in Fig. 9, in exact accordance to a similar previous study for direct comparison. See Sec. 5.3 (L443–483) and the newly added Table 3. Unfortunately, we did not have the time to analyse in depth the dependency with climate variables like temperature and precipitation as suggested, but we would like to point out that such an analysis is better performed with annual mass balance series, not cumulative mass balance as in our study, and has already been carried out, e.g., in Huss et al. (2010)a.

The authors don't need to address all of these questions, but I encourage them to take stock of their hardearned new dataset and explore some of the implications. Second, I thought the description of the estimation and propagation of different sources of uncertainty (section 3) was excellent. It was concise and efficient, yet detailed enough to be replicated. Well done.

We thank the reviewer for this very positive appreciation of our work.

I only have one suggestion, which is that I think the elevation change rate (m/yr) uncertainty for \*each individual glacier\* should be better conveyed to the reader.

Fig 4 has now been improved to more clearly show the total per-glacier elevation change rate uncertainty. The histogram is now cumulative and better shows the uncertainty in relation to the relative frequency of glaciers.

For example, Fig. 4 should have another panel showing a histogram of the cumulative uncertainty (all sources) for each glacier.

Fig. 4C now shows cumulative histograms and includes the total uncertainty for each glacier

Then, the median of this distribution can be reported in the abstract, results, etc. For example, the Swiss-wide mean glacier mass balance was -0.52+/-0.09 m/yr during the period 1931-2016. On average, for an individual glacier, what is the uncertainty in glacier mass balance? Is it 0.1 m/yr, 0.6 m/yr, etc.?

We added the range and the median glacier-specific total specific mass balance uncertainty in Sec. 5.2 (L418): "While our Swiss-wide total specific mass balance uncertainty is low  $(0.09 \text{ m w.e. a}^{-1})$ , uncertainties at individual

glaciers are in the same order of magnitude of (and sometimes higher than) the magnitude of change  $(0.01-0.62 \text{ m w.e. a}^{-1})$ , with a median of  $0.17 \text{ m w.e. a}^{-1}$ ."

I think this is an important number to convey clearly, since it helps the reader understand how much uncertainty is associated with an individual glacier mass balance estimate (rather than the regional and Swiss-wide averages). Overall, this is a very impressive and exciting manuscript, and I recommend publication with minor revisions.

We thank the reviewer for the constructive review and kind words.

**5.1 Specific comments**

Lines 21-22: It might provide useful context for the reader if you plot the glaciological mass balance data for the handful of monitored glaciers in Switzerland and the European Alps—for example, can Fig. 1 include a sub-panel that has a plot like Fig. 5 of Huss et al. (2015) or Fig. 1 in Oerlemans (1994), with the published of field-based glacier change data for European glaciers? I think that plot would effectively convey both how sparse the existing data are (i.e., how few glaciers have repeat field-based monitoring), and a sense of the patterns and magnitude of glacier change over the last century.

Replaced the Fig. 1 precipitation series (which was not mentioned in the text) with a view of the mean annual mass balance data that we used for our temporal standardisation (now Fig. 1D). We also replaced the weather station locations in Fig. 1A with the locations of all mass balance series that we used.

Lines 76-77: "The cameras were of two different brands called "Wild" and "Zeiss", which had different focal lengths, image dimensions, and frame appearances..." Perhaps you should be more specific here—you explain that the cameras are of two different brands, but are there multiple camera types and lenses of each brand? In other words, do you solve for a full separate set of camera parameters for each of the 21 individual cameras used, or, are there camera distortion parameters that are largely shared between the two groups of Wild vs. Zeiss cameras? You mostly answer this question in lines 165-170, but I'm still curious whether you shared any of the camera parameters across different individual cameras of the same model. Alternatively, after the fact, did you explore how different the Metashape-determined camera parameters varied across individual cameras of the same model? That would be an interesting observation to share with folks who want to do similar regional reconstructions for other glaciated areas.

This suggestion added fascinating information on the intrinsics on our analysis, and we thank the reviewer for the suggestion. The variation between camera models of the same camera (more specifically, camera models from subsets with the same camera name; we do not know to what extent they varied mechanically during their lifetimes) was low for the estimated focal length, but significantly higher for all other distortion parameters. We added a description of this: "The discrepancy between camera models of different years is low for the estimated focal length (0.39% standard deviation on average), while all other distortion parameters (radial, decentering, principal point offset, affinity, and skew) have a significantly higher spread of 37% on average. The large spread of the distortion parameters could indicate overfitting, but with acceptable tie point residual errors (Table 1), we choose to not consider this a significant issue." The camera distortion parameters are also available in the supplementary data, in case this may be of interest in the future. This would have been fascinating to study further, for example by simplifying the camera model to reduce the spread, but each photogrammetric run took about three months with six processing machines running in parallel. We unfortunately do not have the opportunity to run this again for this study.

Line 116: "In a nutshell, images and associated metadata are preprocessed first for homogenizing the inputs." – Can you be a little more specific here? What are you homogenizing? Perhaps you should say "...images are preprocessed first to remove the geometric distortions introduced during digitization (scanning) and to correct for biased position data."

Thank you for the suggestion, we applied this change

Lines 135-137: Can you give the reader a little bit of intuition for why, in 32% of the images in the dataset, only 3 fiducial marks could be identified? Were the images too degraded to see the pattern by eye, or was it just that the automated detection couldn't spot the fiducial marks in regions of the images that were very dark, etc.?';

To address this question, we have added in the revision the sentence: "Model mismatches are largely due to film scratches, damages, or poor contrast in the images, and are easier to identify manually than to account for in a more complex model."

Lines 144-149: Can you add a sentence to explain how the initial position data from 1931 was determined? Why does this slight bias exist in the position data?

The biases are present in the position data provided by swisstopo. We have added in the text that the data are directly from swisstopo (the sentence ends with "provided by swisstopo"; see L149 in revised version) to clarify. We have added a sentence to provide a possible explanation: "A possible explanation for these biases is that camera positions were estimated by triangulation from reference points, which in turn were all positioned relatively to other reference points. Therefore a small positional error could accumulate as the point is further away from the main reference." (L 155)

Lines 167-170: I'm interested and a little surprised that the same individual camera can have different parameters (as constrained during the Metashape reconstruction) from different years! How much do the distortion parameters vary? Is it more than you expect from the uncertainties associated with the Metashape reconstructions? Have you done any sensitivity experiments to make sure that you're not overfitting the camera parameters in Metashape due to the inherent limitations of the dataset—minimal spatial coverage, physical image degradation, poor overlap, etc.?

We consider this solved together with the "Lines 76–77" comment reply. (Text has been added for camera model spread on L180)

Lines 208-210: Can you give the reader a little intuition for the reason of the limited glacier coverage (55% of the glaciated area is missing). Is this largely because much of the glacier area was not imaged from 2 viewing angles (due to occlusion and a limited number of photograph stations?), or, is it that much of the glacier surface didn't have enough "features" (texture) that could be matched during the photogrammetric reconstruction? If it's the latter, it seems surprising that the upper-elevation areas, with more snow/firn cover, wouldn't have sparser coverage?

We have added a brief explanation for the gaps in the Methods section "(due to incomplete image coverage and processing shortcomings)" L XX and the reasons are further detailed in the discussions section 5.2 (L 422-429).

Lines 321-322: Can you put this number in context? What percent of the total eustatic sea level rise (1931-2016) is represented by the melting of Swiss glaciers?

Added sea level equivalent in the second sentence of the Results (L 341).

Lines 331-334: Perhaps you can address this in the discussion rather than the results, but I think you should add a sentence or two to explain what you think the mechanism is to explain why ice at higher elevation is experiencing less negative mass balance, and whether you think this trend is going to be strengthened or weakened in the future.

This is a good point. We have now added an explanation for the correlation at the end of section 5.3. "We attribute the correlation between median elevation and the rate of glacier mass loss to differences in snow accumulation. As air temperature forcing is similar throughout the Swiss Alps, glaciers at higher elevation are characterised by less snow precipitation that leads to smaller mass balance gradients, and hence, a smaller sensitivity to air temperature change (Fischer et al., 2015)."

Lines 409-410: "The uncertainty analyses of this study show that the regional aggregates are accurate, thus indicating that a reanalysis of the TerrA imagery with modern methods may be worthwhile especially if interested in individual glaciers." See my second general comment in the first paragraph of this review. I think it would help the reader to convey more clearly what the uncertainty is for the mass balance estimate of any individual glacier. This could be conveyed effectively with a histogram where the x-axis is uncertainty (m/yr) and the y-axis is the number of glaciers (like Fig. 4C, but showing cumulative uncertainties).

Fig. 4C has now been changed to show a cumulative histogram with the total glacier-wise uncertainty.

Figure 1: Perhaps you should include a small inset in Fig. 1A to show the map of all European glaciers and country borders, with a rectangle highlighting your study area shown in (A)?

Added an inset in Fig. 1 (now Fig. 1B) highlighting the location of Switzerland in Europe. Adding its glaciers was difficult in the limited space, so we opted to only have country and coastal borders.

Figure 4: I think the most useful error-related figure for the reader would be a modification of (C) in which you plot the histogram of the \*total\* elevation change rate uncertainty (propagated from all the different components of uncertainty shown as separate colors in Fig. 4C) for the glaciers you reconstructed. What's the median glacier-specific dH/dt uncertainty? That would be a really helpful statistic to report widely in the paper (e.g., in the Fig. 4 caption, in the results, and in the abstract and conclusion).

We changed panel C in the figure to include the total uncertainty. In addition, the histograms are now cumulative histograms shown as lines, making it easier to visually parse what is most significant. In Sec 5.2 (L 420), the median total glacier-specific mass balance uncertainty is now specified: " $(0.01-0.62 \text{ m w.e. a}^{-1})$ , with a median of  $0.17 \text{ m w.e. a}^{-1}$ )"

**5.2 Technical comments/corrections (line edits)**

**Line 40: remove "that is" We applied this change**

Line 45: "of nearly all glaciers in Switzerland" – can you finish this sentence with the number of reconstructed glaciers in parentheses? (i.e. "... of nearly all glaciers in Switzerland (XXX)").

Added "(89% by count)"

**Line 115: remove "being" We applied this change**

There's a mix of British- vs. American-style spelling of various words—I'd stick to one or the other for consistency. It seems that the American style is used most of the time. Here are a few occurrences of British English that I saw:

- Line 46: centre  $\rightarrow$  center
- Line 100: centre  $\rightarrow$  center
- Line 125: centrepoint  $\rightarrow$  centerpoint
- Line 193: centre  $\rightarrow$  center
- Line 327: analyse  $\rightarrow$  analyze
- Line 331: Analysing  $\rightarrow$  Analyzing
- Line 401: analysed  $\rightarrow$  analyzed

We choose to be consistent with British English.

Line 129: replace "increase" with "introduce" (since 2 fiducial marks represents zero redundancy for the similarity transformation, so 3 fiducial marks introduces redundancy). We applied this change

Line 342: replace "and based" with "or based." We applied this change

Line 396: remove "sheer" We applied this change

Line 396: replace "Whilst" with "Although" We applied this change Line 405: "improving"  $\rightarrow$  "improve" We applied this change

Line 406: "pave the way to an increased coverage when compared to..."  $\rightarrow$  "pave the way to increased coverage compared to..."

We applied this change

Line 411: "or as alluded to above"  $\rightarrow$  This phrase is super vague—an easy fix would be to use parentheses to cite a handful of other examples of using historical imagery to reconstruct glacier change.

Added all references that were "alluded" to!

Line 419: should "database" be "databases"? I think this word should be plural. Changed "database" to "databases".

Line 432: add commas after "approach" and "ago" We applied this change

**6 Referee comments 3 (RC3)**

Review for "Halving of Swiss glacier volume since 1931 observed from terrestrial image photogrammetry" Authors: Erik S. Mannerfelt, Amaury Dehecq, Romain Hugonnet, Elias Hodel, Mattias Huss, Andrea Bauder, Daniel Farinotti General Comments: Study uses tens of thousands of terrestrial photographs collected from 85% of the glaciers located in Switzerland to generate historical surface elevations and determine 85-year mass balance change. The amount of work performed in this study is extraordinary and the results are very exciting. The methodology used by the authors for elevation derivation, calculating mass balance, and estimating error is sound. The majority of my comments in the manuscript are to do with seeking a bit more clarification in the text.

We thank referee RC3 for their very positive comments on our work and particularly their time to help us improve the manuscript presentation. In the following part, we address their comments.

"Historical" and "Historic" are used throughout. "Historic" is subjective on the authors' part, so the more appropriate adjective is "historical".

Changed all occurrences of "historic" to "historical" throughout the text.

The word "taken" in referring to image acquisition is colloquial. It is better to use "acquire" or "collect" instead.

Replaced all occurrences of "taken" (in regards to photographs) with "acquired"

In Section 2.3, because glacier outlines are being derived from historical topographic maps and datums/ellipsoids evolved constantly throughout the timeline of this study, it is necessary to mention whether digitized versions of the Siegfried maps and the LK50 maps are currently available in their original horizontal projections, or if transformations were preformed so they align with the more current datasets like the swissALTI3D DEM.

Added "(in the Swiss CH1903+ / LV95 coordinate system; EPSG:2056) provided by swisstopo" to the first sentence in Sec. 3.1.3. We don't know the exact specifics of the georeferencing, but these uncertainties will implicitly be included in the uncertainty assessment because we evaluate the spatial discrepancy between our sparse glacier outlines and the outlines derived from the map data, so it will not change the final result.

Admittedly, I am not that familiar with the historical Swiss topographic series, but I am assuming that a different datum was used in its assembly than what is in practice today. This is certainly the case with the American 7.5 minute series topo maps, where XYZ values derived using NAVD29 have a horizontal shift of 400-500 m for regions of greater topographic terrain variability. For the sake of transparency of potential error sources, it is necessary to at least mention whether all datasets were transformed to the same map projection. With several recent studies using datasets comprised of multi-temporal acquisitions (Rignot et. al., (2021); Haubner et. al., 2018; De Rydt et. al., 2021; etc.) as discrete timestamps, I very much appreciate the authors' efforts to standardize the historical elevations' timeline. I wish more studies were as arduous and feel this paper is an example of "best practices" that I hope future multi-temporal glaciological research would benefit by following.

We thank the reviewer for the very positive appreciation and similarly hope that such methodology will benefit future studies.

The one addition that I request be included with the temporal error assessment is to show the absolute differences with a sample of dh results from specific timestamps to the standardized timeline.

Thank you for the suggestion but we believe that the reviewer's suggestion is either already satisfied or cannot be satisfied. On Figure 10, we already compare the modelled and observed cumulative mass balance for the same time periods and for the few modelled glaciers. Additionally, we are not so concerned about the absolute mass balance values (which are not known for each individual glacier), instead what we use is the ratio in cumulative mass balance between two periods (t0-t1 and 1931-2016). We are unable to validate these ratios even for sampled glaciers from our existing observations as this would require having at least two time periods of observation, while we have only one. Please let us know if we have misunderstood your suggestions.

The state of the manuscript is that of a methodology paper. If that is the intent of the authors, then ignore this next part. However, the impact of this paper could be significantly improved if in the discussion section you were to include the broader impacts of your mass balance results with topics like future predictions of Swiss glacier volumes, future melt-water flux to stream discharge rates, anthropomorphic impacts from 1916-2016, etc.

Indeed, the paper is more focused on the methodological developments and the new data set. But following the reviewer's suggestions we have decided to add two subsections in the discussion (Sec. 5.3 and 5.4) where we elaborate both on comparisons to other studies, and on potential glaciological explanations for our findings. Further, we added text on what these data can be used for in the future (the latter half of Sec. 5.4), of which future predictions, discharge, and anthropogenic impacts are excellent suggestions.

You've achieved an amazing temporal scale for mass balance changes over such a large spatial extent that it would be beneficial to the glaciological community if you could include its relevance in a broader scope.

We thank the reviewer for their kind words. With the new addition of further discussion on the implications (Sec. 5.3 and 5.4) and global sea level equivalent contextualisation (L 341), we now believe that we have emphasised the broader relevance further.

**6.1 Specific Line Comments:**

Line 8: I think you mean "conducive" rather than "conductive" We changed to "conducive"

Line 20: "understand" is odd in this context. "resolve" would be a more appropriate word. We changed to "resolve"

Line 33: Does Switzerland refer to the country or its glaciers? Replaced "Switzerland" with "Swiss glaciers"

Line 40: does "photographed position" refer to the camera's position or the image's full extent? Replaced "photographed position" with "camera position" Line 44: using the term "re-process" is confusing. I am assuming that you are referring to the processing that was done with these images to generate the original topo maps in which case it would be better to say "digitally process" instead of "re-process".

We changed to "digitally process"

Line 53: Replace "houses" with "contains" We applied this change

Line 56: Delete "of" We applied this change

Line 69: Replace "will be" with "is" We applied this change

Line 76: Replace "5" with "five" for continuity We applied this change

Line 77: It is unclear exactly what "frame appearances" means. Maybe "frame geometry" would be better? Changed "frame appearances" to "frame geometry"

Line 126: Can you go into more details about the similarity transformation and how it is performed?

It is a generic call to the Similarity constructor in the "scikit-image" Python package, which uses least-squares to find the optimal transformation. The name of the package is now added in the text.

Line 129: Is "marks" referring to fiducial marks?

Replaced all instances of "marks" with "fiducial marks" in the sentence

Line 132: "Manual data" is awkward; is the model that is being referenced the same as the previous sentence?

Replaced "manual data" with "manual fiducial mark identifications"

Line 137: "manually mark" is unclear. Do you mean "identify"? Changed "manually mark" to "identify"

Line 144: Is the metadata digital or analog? Where do the image positions come from?

Added the keyword "digitised" to show that we were given digital versions. And added that the location data was "provided by swisstopo". We were simply given the data without an exact explanation of how the locations were calculated.

Line 150-151: Verb tense is in the past. Also, the difference value being reported, is that a vertical difference? Can you also provide the horizontal difference?

The entire paragraph is now changed to be in present tense. We added the horizontal differences before and after the correction: "[...] and an average horizontal offset of  $0.66\pm4.49$  m" (before), "[...]  $0.70\pm5.18$  m ( $0.39\pm4.15$  m horizontally)." (after)

Line 152: Did you perform the map digitization? If not, then the heading should reflect generating glacier outlines instead.

Changed the title to say "LK50 map series glacier outline digitisation"

Lines 153-163: Please include somewhere in this section what the average glacier area is to provide scaled-context for researchers primarily doing work in Greenland and Antarctica.

We added glacier count and mean area on L 167: "[...] modifying the 2,491 outlines (with a mean area of 0.60 km2) [...]"

Line 153: "digitizing the scanned and georeferenced LK50 map series." It is unclear here whether the maps are being digitized or the glacier outlines.

Clarified to say "[...] digitizing outlines on the scanned and georeferenced LK50 map series."

Lines 154-155: The verbs in this sentence are written in the past tense. Also, use "sporadic" or "inconsistent" rather than "spotty"

Replaced "spotty" with "sporadic"

Lines 161-161: The verb in this sentence are written in the past tense.

We changed to present tense

Lines 165-166: Suggested rewrite for "version 1.6.5 by dividing the images in subsets, and processing each subset individually (Table 1)." with "version 1.6.5 in separate subsets (Table 1)." We applied this change

Line 166: Replace "one" with "each", "taken" with "acquired", "certain" with "specific", and "at a certain" with "over a single".

Applied all suggested changes

Line 167: Should "groups" be "subsets"? If not, then you need to define what a "group" is. Changed "groups" to "subsets".

Lines 167-168: Replace "subset, as the used cameras might vary in their distortions parameters, depending on their construction and in which year they were used;" with "subset because of varying distortion parameters for each camera;"

We applied this change

169-170: Do you have evidence to support this statement? It really should have a citation or maybe documentation from historical metadata.

"May" is the keyword here. To our knowledge, there is unfortunately no information about the exact histories of the cameras, so we can only speculate.

Line 171: Are stereo-panoramas one per subset? Are these the camera models generated in the previous paragraph? If so, then say something like "the computed camera-models" so there is no question that potentially new camera-models have been introduced.

Replaced "merged using the same camera model" with "merged with a recomputed camera model" to clarify. The first question is arguably already accounted for; in the data description, it is said that each stereo-panorama consists of 4–5 images, and in the paragraph before, that 192 images on average make up a subset.

Line 172: It is unclear what "others" is specifically referring to. Is it other "panoramas" and/or is it other image subsets? This also applies to "them" on Line 174.

Changed "others" and "them" to "panoramas" and "stereo-panoramas", respectively

Line 177: In referring to the co-registration, you need to define what is being co-registered in regards to stable terrain or over entire generated data coverage.

Clarified by adding "[...] all areas where the dense clouds overlap,"

Lines 180-187: In determining the conversion of DEM grids from 3D point clouds, how did you decide on the cell size? What did you determine for a cell size? Was the entirety of each point cloud utilized in DEM conversion? Studies using oblique aerial imagery have found that there is a distance from point of image acquisition that the generated elevations are no longer reliable (Girod et. al., 2018); was this phenomenon a factor with the terrestrial imagery? If so, was this consistent across all historical acquisitions?

The grid size was decided qualitatively based on an initial test of only one glacier. We discussed coarser resolutions, but opted to not sacrifice on detail where it exists (since the actual resolution varies). We did not filter the points based on distance to the camera, but this is inherently accounted for in part; the point density will reduce with distance, meaning that overlapping point clouds will have a natural averaging weight due to the differential point spacing. In addition, we visually observed that distant points had a low proprietary Metashape "confidence" score (c.f. L 195) and most distant points were excluded in this filtering step. We

observed no clear difference in the reliability of reconstructed points with increasing distance in relation to the capture date. The images are generally of equal quality over the entire study period.

Line 180: Is the stereo-panorama alignment being applied to all panoramas generated from the 21,000 images or to just panoramas from a single subset?

Only within a subset ("within a subset" is in the sentence.). This was a decision based on the logistics of running this analysis on six different machines in parallel; each subset needed to be independent for the task scheduler. Since the subsets generally do not overlap, however, this should not have a considerable effect on the outcome.

Lines 182-183: It is untrue that details for the confidence parameters are not available. White papers written by Agisoft Metashape are readily available online that describe how the confidence parameter is determined based on the contributing depth maps. Simply reporting a value of "2" without any supporting evidence to validate its use systematically appropriate is imprecise—please amend this.

Thank you for the suggestion but we have not been able to find the mentioned white papers. Would you mind providing a link to these resources? Without this information, we are unable to address the question. Regarding the confidence threshold, this was chosen qualitatively. A confidence threshold of 1 was still very visibly noisy, while a threshold of 3 excluded points that visually looked reliable. We thus settled on 2. We added a sentence to clarify this right after the confidence is introduced on L 194: "This threshold is chosen qualitatively by visually assessing the effect on noise for a few arbitrarily selected dense clouds."

Lines 183-184: It is unclear what relevance draping the orthoimages over the DEMs has for the processing and I recommend deleting "and after orthoimages are draped on the DEMs," from the sentence.

It was indeed unclear what this part meant. Changed to " and after orthoimages are generated using the DEMs". The part cannot be deleted because orthoimage generation is otherwise not mentioned.

Line 186: Alter "position and rotation" with "positional and rotational errors" We applied this change

Line 187: Include "historical" before "DEMs". We applied this change

Line 189: I suggest calling the "correction" an "adjustment" or "standardizing" because time itself is not in error.

Consistently changed to "standardising"

Lines 190-192: The parameters described in the first two sentences have already been explained previously in the manuscript. You are not technically correcting the temporal scale, but instead standardizing it to specific years. It would be better in this paragraph to explain why the temporal scale requires standardizing, the specific timeline of standardization for each dataset, and the potential errors that may arise when rates are not based on the absolute temporal scale.

You are correct that this is a repetition. We have shortened the text to avoid this long repetition: "Because both the historical and modern elevation data set were acquired over several years (1916–1947 and 2007–2018, respectively), it is necessary to standardise the periods of observation for facilitating evaluation and interpretation of the results." We have replaced the term "correct" by "standardise". We do not elaborate on the importance of the time standardisation here, but this is discussed in detail in sections 5.2 and newly added 5.4.

Line 193: Alter "correct" with "adjust" We applied this change

Line 195: Delete "anomalies in" because you are using the full extent of mass balance calculations and not just anomalies.

Indeed, we use mass balance "anomalies" from a reference period in our approach to temporally homogenise the volume changes. So this addition cannot be removed. But we have clarified that part of the text: "Regional anomalies in mass balance from the reference period 1961–1990 have been derived [...]". Line 198: It is unclear if "respective region" refers to each of the four glaciers separately or all of them lumped together.

We reformulated on L 209-210: "The difference in cumulative annual mass balance for the region hosting the respective glacier between the reference period of 1931–2016 is compared [...]"

Line 199: It is unclear with the "factor" is.

We changed to "fractional difference"

Line 200: Alter "corrected" with "standardized" We changed to "standardised"

Line 205: Alter "corrected" with "standardized" We changed to "standardised"

Line 207: State whether the percentage of missing data is the result of image coverage extent or processing, or both.

Added "(due to incomplete image coverage and processing shortcomings)"

Line 208: Delete "the whole" We applied this change

Line 209: Can you quantify "significant trend"? What is the trend in reference to? Verb use is in past tense.

This sentence was a convoluted way to say "not all values portray the terminus; there are values in the accumulation area too. Therefore, we chose a hypsometric interpolation approach". The "significant trend" is now replaced with "throughout their elevation ranges"

Line 212: Referencing "xdem" needs a citation or a reference that it is a python module. This is already established on line 176

Added "[...] Python package [...]" to the sentence.

Line 214: What is the "threshold" in reference to? This needs more details and if the threshold is based on other research, a citation as well.

Rephrased the sentence to be clearer and to not use "threshold"

Line 215: Delete "the" We applied this change

Line 216: Are full data gaps replaced with the single value or is there a maximum distance where new values are not assigned when using this interpolation technique? Please include either scenario in the text.

As in the global hypsometric interpolation method of McNabb et al. (2019), we replace all gaps by the mean value in the corresponding elevation band. Therefore there is no filtering based on maximum distance. This is now clarified in the revised text L223-230.

Line 218: Recommend altering "each entry of the SGI2016 individually." with "each SGI2016 outline." We applied this change

Line 221: Alter "historic" with "historical" We applied this change

Line 236: What are the "elevation measurements"? Are these the generated historical elevations? If so, then say "stochastic processed elevation errors" instead because "elevation measurements" sounds like altitudes collected in situ. Otherwise, state where these measurement come from.

We changed to "stochastic elevation change measurement errors"

Line 237: Alter "corrected" with "standardized"

Renamed "temporal correction" to "temporal standardisation" in all occurrences in the text

Line 239: Alter "presented through" with "within" We applied this change

Line 240: Suggested rewrite of "from the scale of pixels to that of glacier, and that of glaciers to regional scales," with, "from scales of pixel to glacier and glacier to region," We applied this change

Line 250: For the stable terrain, do you include all surface slopes as well?

We used all surface slopes because we found no significant correlation between slope and elevation change uncertainty (written in section 3.5.1)

Lines 257 and 258: Phrases "might be affected" and "might cancel out" are too uncertain and do not support your argument. Either provide a reference to past research who has identified similar results, or a statistical assessment to be able to assert that the quantity of DEMs in this study is indeed the reason DEM differencing is unaffected by terrain parameters like slope. Also, "these" on Line 258 referring to the attribute-dependent errors?

This is true that our explanation is only speculative. However, understanding the reasons why the error does not correlate with terrain attribute would not impact our interpretation of the results and hence we decided not to spend more time on this issue.

Lines 270-272: What about the maps' error? There absolutely was error in the in situ triangulation estimates and original photogrammetric work—are there no estimates of the actual maps' accuracy?

The error of the map is already mentioned: "the georeferencing error of the map" and "errors in manual delineation [...] of a glacier front"

Line 272: Are there no references as to the years of data collection by map? Do the maps not contain this information? Were any reports written with the topo maps? US and UK topo maps contain temporal information and often had supplemental reports written to validate methods used when compiling the maps.

Added "(which is unknown, [...]" in the beginning of the parenthesis to clarify.

Line 273: It is unclear what "degree" is specifically referring to.

We changed to "[...] as the *extent* of which data [...]"

298: "Polluted" is an odd word choice. I recommend using something like "biased" instead. We changed to "biased".

Line 301: It's annoyingly pedantic, but "small scale" is actually referencing a very large extent. For continuity with the rest of the text, it would be better to reference either "pixel-scale" or "glacier-scale".

Good point! Changed to "pixel scale"

Lines 302-303: "largely independent among glaciers of a given region." Does this mean the errors are independent by glacier, or by region?

Replaced "from" with "between" to clarify that uncertainties are largely independent between glaciers

Line 305: Recommend altering "correction" to either "adjustment" or "standardized" Renamed "temporal correction" to "temporal standardisation" in all occurrences in the text

Line 307: Is this uncertainty based on mass balance derived from remote sensing methods? If so, that should be stated.

This uncertainty estimate refers to the determination of annual glacier mass balance based on the glaciological method, i.e. in situ, resulting from all individual components of the uncertainty. The estimate provided by Zemp et al. (2013) corresponds to a consensus for the most important monitoring programmes in Europe. To clarify this the text has been updated as follows: "The reported uncertainty for one year's specific mass balance ( $\sigma_B$ ) based on in situ glaciological observations is approximately 0.2 m w.e.  $a^{-1}$  (Zemp et al., 2013)."

Line 321: The minus symbol is unnecessary because you say there is a "loss". Removed the negative sign

Line 323: For continuity in the text, please also provide the 1931 glacier volume.

The Results sentences have already been elaborated with additional numeric information. For the sake of readability, we opt to not add additional numbers in the text (to preserve readability), since this information is already available in Table 2.

Line 341: Are these "glaciers" also Swiss? Please provide a regional reference to the glaciers.

Added "Swiss"

Line 351-353: This sentence is difficult to understand, particularly the portion on Line 353.

Thanks for making us aware of this. The sentence has been reformulated as follows: "The uncertainty in annual mass balance provided by this dataset is mainly given by the precision of the repeated DEMs used for constraining long-term mass change. The mass balance model, which provides annual variations that are consistent with these mass change observations, only marginally contributes to the overall uncertainty."

Line 354: What specifically are the "components" referring to?

Reformulated and clarified: "... allowing for conservative estimates of the uncertainty components that are difficult to be inferred retrospectively."

Line 356: Suggest altering "when considering the entire period of interest." with "for the study's timeline." We applied this change

Line 356: Suggest altering "The comparison to results of the present study" with "Comparing those results to this study's".

We applied this change

Line 358: Replace "smaller" and "study" with "less" and "estimates" respectively. We applied this change

Line 359: What year is the map data from? Is it the same as the imagery? Was the map data and the derived historical DEM data referenced in the same vertical datum? If they weren't, then what transformation was used with the map data?

Added "in the same time-intervals" in the caption of Fig. 10A. The vertical datum is CH1903+, which is now specified in Sec. 3.1.3: " (in the Swiss CH1903+ / LV95 coordinate system; EPSG:2056)"

Line 405: Alter "improving" to "improve"; is "description" the word you meant here? If so, it is unclear what you are specifically referring to.

Changed "improving" to "improve". "Feature description" is the terminology for numerically parameterising (or describing) arbitrary features in an image for example used by opency (https://opencv.org). SIFT and ORB are arguably the most heard of feature descriptors. They are sometimes erroneously called "feature matchers", but matching is a separate step after description.

Line 432: Include commas after "approach" and "ago"; verb tense in this sentence is past. We applied this change.

Line 446: "high potential" is awkward.

Reworded to "[...] equal potential for evaluating other historical datasets [...]"

**References**

De Rydt, Jan, Ronja Reese, Fernando S. Paolo, and G. Hilmar Gudmundsson. "Drivers of Pine Island Glacier speed-up between 1996 and 2016." The Cryosphere 15, no. 1 (2021): 113-132.

Dussaillant, I., Berthier, E., Brun, F., Masiokas, M., Hugonnet, R., Favier, V., Rabatel, A., Pitte, P., Ruiz, L., 2019. Two decades of glacier mass loss along the Andes. Nat. Geosci. 12, 802–808. https://doi.org/10.1038/s41561-019-0432-5

Girod, Luc, Niels Ivar Nielsen, Frédérique Couderette, Christopher Nuth, and Andreas Kääb. "Precise DEM extraction from Svalbard using 1936 high oblique imagery." Geoscientific Instrumentation, Methods and Data Systems 7, no. 4 (2018): 277-288.

Haubner, Konstanze, Jason Box, Nicole Schlegel, Eric Y. Larour, Mathieu Morlighem, Anne Solgaard, Kristian K. Kjeldsen et al. "Simulating ice thickness and velocity evolution of Upernavik Isstrøm 1849-2017 with ISSM." In AGU Fall Meeting Abstracts, vol. 2017, pp. C11B-0911. 2017.

Huss, Matthias, Laurie Dhulst, and Andreas Bauder. "New long-term mass-balance series for the Swiss Alps." Journal of Glaciology, 61.227 (2015): 551-562.

Oerlemans, Johannes. "Quantifying global warming from the retreat of glaciers." Science, 264.5156 (1994): 243-245.

Rignot, Eric, Lu An, Nolwenn Chauche, Mathieu Morlighem, Seongsu Jeong, Michael Wood, Jeremie Mouginot et al. "Retreat of Humboldt Gletscher, north Greenland, driven by undercutting from a warmer ocean." Geophysical research letters 48, no. 6 (2021): e2020GL091342.